# Layer-Wise Modality Decomposition for Interpretable Multimodal Sensor Fusion

**Jaehyun Park**    **Konyul Park**    **Daehun Kim**    **Junseo Park**    **Jun Won Choi**[*]

Seoul National University

{jhpark, kypark, dhkim, jspark}@adr.snu.ac.kr

junwchoi@snu.ac.kr

## Abstract

In autonomous driving, transparency in the decision-making of perception models is critical, as even a single misperception can be catastrophic. Yet with multi-sensor inputs, it is difficult to determine how each modality contributes to a prediction because sensor information becomes entangled within the fusion network. We introduce Layer-Wise Modality Decomposition (LMD), a post-hoc, model-agnostic interpretability method that disentangles modality-specific information across all layers of a pretrained fusion model. To our knowledge, LMD is the first approach to attribute the predictions of a perception model to individual input modalities in a sensor-fusion system for autonomous driving. We evaluate LMD on pretrained fusion models under camera–radar, camera–LiDAR, and camera–radar–LiDAR settings for autonomous driving. Its effectiveness is validated using structured perturbation-based metrics and modality-wise visual decompositions, demonstrating practical applicability to interpreting high-capacity multimodal architectures. Code is available at `https://github.com/detxter-jvb/Layer-Wise-Modality-Decomposition`.

## 1   Introduction

There is a growing move toward perception models that leverage multimodal inputs to improve downstream task performance. In autonomous driving, for example, modern systems increasingly fuse diverse modalities to produce more accurate and contextually relevant outputs. Incorporating different sensor streams such as camera, radar, and LiDAR has consistently improved reliability. Yet this integration complicates interpretability and transparency: when the model makes a decision, it becomes hard to discern to what extent each modality contributed to the final outcome.

This challenge is particularly pronounced in safety-critical domains such as autonomous driving, where model outputs directly affect safety and reliability. Consequently, improving the interpretability of multimodal perception models is essential. Although interpretable AI has advanced in various domains, its application to autonomous driving remains underexplored. To bridge this gap, it is necessary to explore how existing interpretability methods can be applied to sensor-fusion models.

Current work in interpretable AI can be grouped into two categories: intrinsically interpretable models and post-hoc interpretation methods. Intrinsically interpretable models have been widely adopted to directly interpret input feature attributions to model predictions. Generalized Additive Models (GAMs) [1] represent outputs as sums of feature-wise functions, ensuring intrinsic interpretability. Recent variants [2, 3, 4, 5] improved flexibility while preserving this property. However, they require structural constraints, which make it difficult to model complex patterns. This inter-

---

[*]Corresponding author.

39th Conference on Neural Information Processing Systems (NeurIPS 2025).

pretability–accuracy trade-off [6] limits their applicability to multimodal perception tasks, where capturing cross-modal interactions is essential.

Post-hoc interpretation methods are particularly advantageous for high-capacity models, as they maintain predictive accuracy without necessitating architectural modifications. Local surrogate models, such as LIME and LORE [7, 8], enhance interpretability by approximating complex decision boundaries with low-capacity models. However, this simplification limits their ability to capture the intricate, high-dimensional dependencies inherent in large-scale perception networks. Global interpretation techniques, such as functional ANOVA (fANOVA) [9], are effective for tabular or low-dimensional data, but they encounter substantial challenges in scalability and decomposition fidelity when applied to multimodal perception models.

Consequently, there is a need for a post-hoc interpretability method that can deliver layer-wise, modality-specific attributions for high-capacity models without compromising performance or requiring architectural modifications. In this study, we propose Layer-Wise Modality Decomposition (LMD), a framework that enables the interpretation of complex multimodal perception models while fully preserving their original performance.

Our approach locally linearizes neural network operations to separate modality-specific information while forwarding it through the model to the final prediction. In particular, we formulate LMD within the Layer-Wise Relevance Propagation (LRP) [10, 11, 12, 13, 14, 15, 16, 17] and Deep Taylor Decomposition (DTD) [18] frameworks, which are widely adopted post-hoc methods in Explainable AI (XAI).

Our LMD is grounded in a rigorous theoretical framework that separates modality-specific components at each layer. We validate its effectiveness on multi-sensor perception models using the widely adopted nuScenes benchmark [19]. More broadly, LMD is generalizable and can be readily applied to other multimodal architectures.

Our key contributions are as follows:

1. We propose LMD, a novel post-hoc interpretability method that decomposes the contribution of each sensor modality across all layers of a complex multimodal model. The decomposition process operates without any modification to the original architecture.

2. We conduct both extensive qualitative and quantitative experiments to validate the proposed method. We introduce novel perturbation-based metrics to assess interpretability of our model.

3. We investigate several LMD variants employing different bias-splitting strategies and assess their effectiveness from a modality separation perspective.

## 2 Related Work

### 2.1 Interpretation Methods

Interpretable methods aim to provide transparency in model predictions by designing architectures in which feature contributions are explicitly traceable. Such models are inherently understandable, removing the need for external post-hoc explanations. Broadly, interpretable approaches can be grouped into two categories: intrinsically interpretable models and post-hoc interpretation methods.

Intrinsically interpretable models—including Generalized Additive Models (GAMs) [1], Neural Additive Models (NAMs) [2], NODE-GAM [4], and Gaussian-NAM [5]—represent model outputs as the sum of feature-wise functions. This additive formulation enables fine-grained attribution and clearer interpretation of individual feature effects. However, the structural constraints that ensure interpretability also limit these models' capacity to capture complex feature interactions and contextual dependencies.

Post-hoc interpretation methods aim to explain the behavior of pretrained black-box models. The methods such as Functional ANOVA (fANOVA) [9] decompose the output variance of a trained model into contributions from individual features and their interactions. While fANOVA is effective for tabular or low-dimensional data, it faces scalability challenges in high-dimensional settings, where decomposed features may become unidentifiable [20].

## 2.2 Post-hoc Explanation Methods

Post-hoc explanation methods aim to interpret complex models without modifying their original architecture. By operating directly on pretrained black-box models, these approaches preserve predictive accuracy while providing interpretability.

Backpropagation-based methods generate explanations by leveraging gradients or gradient-related signals within the model. For instance, Layer-Wise Relevance Propagation (LRP) [10] assigns relevance scores to input variables by decomposing layer-wise computations, thereby producing attention maps that can be interpreted as input-level contributions. The propagation rules in LRP are grounded in the Deep Taylor Decomposition (DTD) framework [18], which requires local linearization of each layer to remove non-linear biases. Building on this foundation, subsequent studies have proposed refined attribution schemes to ensure faithful relevance propagation across diverse network components such as residual connections, attention mechanisms, and normalization layers in modern deep architectures [11, 15, 13, 14, 17, 16, 12].

## 2.3 Multi-sensor Perception

Multi-modality perception leverages the complementary strengths of different modalities to achieve robust scene understanding in autonomous driving. Cameras offer rich semantic context, LiDAR provides precise 3D geometric information, and radar ensures reliable velocity and range estimation under adverse conditions. To harness these complementary signals, recent models explore intermediate fusion strategies that align features from different modalities [21, 22, 23, 24, 25].

For instance, BEVFusion [22] employs modality-specific encoders for camera and LiDAR, followed by fusion via spatially consistent representations. RCBEVDet [25] introduces a radar-specific encoder and integrates its output with camera features using deformable cross-attention [26], facilitating effective cross-modal alignment.

Despite their strong performance, these models often lack interpretability. During the fusion process, features from different sensors are aggregated, making it difficult to disentangle the contribution of each modality to the final prediction. This lack of transparency complicates failure diagnosis and system validation, highlighting the need for interpretable techniques in safety-critical applications.

## 3 Background

In this section, we briefly review the Deep Taylor Decomposition (DTD) [18], and the Layer-Wise Relevance Propagation (LRP) [10] before introducing our LMD method.

Let $\mathbf{f} : \mathbb{R}^N \to \mathbb{R}^M$ be a vector-valued function consisting of a deep network with the input neurons $\mathbf{x} = \{x_i\}_{i=1}^N$. Let $f_j$ be the $j$-th output neuron. DTD decomposes $f_j$ into contributions from $N$ input variables using first-order Taylor expansion of $f_j(x)$

$$f_j(\mathbf{x}) = f_j(\tilde{\mathbf{x}}) + \sum_i \mathbf{J}_{ji} (x_i - \tilde{x}_i) + \epsilon = \sum_i \mathbf{J}_{ji} x_i + \underbrace{f_j(\tilde{\mathbf{x}}) - \sum_i \mathbf{J}_{ji} \tilde{x}_i + \epsilon}_{\text{bias } \tilde{b}_j}, \tag{1}$$

where $\tilde{\mathbf{x}}$ is a reference point, $\mathbf{J}_{ji} = \frac{\partial f_j}{\partial x_i}\big|_{\mathbf{x}=\tilde{\mathbf{x}}}$ is the Jacobian at $\tilde{\mathbf{x}}$ and $\epsilon$ is the first-order approximation error. The summation term containing $x_i$ represents the first-order contributions of input variables to the output $f_j(\mathbf{x})$. With $\tilde{\mathbf{x}}$ treated as a constant, the bias $\tilde{b}_j$ consists of constant terms and $\epsilon$, which represents high-order interactions determined by the order of differentiability of the function along the line segment joining $\mathbf{x}$ and $\tilde{\mathbf{x}}$.

Prior LRP studies [18, 10, 14, 13, 17, 11, 15] developed local linearizations of non-linear network layers to attribute predictions to input variables. Within the DTD framework, two core design choices arise: (i) selecting the root point at which the local Jacobian $\mathbf{J}_{ji}$ is computed, and (ii) specifying how the bias term is redistributed to the inputs. For element-wise non-linear activations, $\mathbf{J}_{ji}$ was typically set to the identity [11, 13] to enforce the conservation property—i.e., that the sum of input attributions is preserved across layers. For LayerNorm, prior work notes that the true Jacobian entries are near zero, rendering direct linearization uninformative [11, 13]. A common workaround treats LayerNorm as effectively element-wise by assuming (local) variance to be constant.

Bias handling has been central to maintaining conservation and differentiates LRP variants. For LayerNorm or other local renormalization operations, some methods distributed the bias uniformly over inputs [14, 15], whereas others applied the identity rule and excluded the bias from redistribution [17, 13]. An alternative was to omit the bias altogether, as [11] argues, based on observed numerical instabilities in prior schemes.

# 4    Method

In this section, we describe the details of the proposed LMD method. The overall process of LMD is depicted in Figure 1. For clarity, we derive the formulation for two modalities (e.g., camera and radar) without loss of generality; the extension to an $M$-modality setting is provided in Appendix F.

## 4.1    Formulation of Layer-Wise Modality Decomposition

We first show that, using first-order Taylor approximation, the features of a fusion model at each layer can be decomposed into modality-specific terms. Let $f^1, \ldots, f^N$ be a set of $N$ layer functions consisting of the camera-radar sensor fusion model performing perception tasks. Suppose the composition of layer functions from the first layer to the $l$-th layer is given by

$$F^l(\mathbf{x_c}, \mathbf{x_r}) = f^l \circ \cdots \circ f^2 \circ f^1(\mathbf{x_c}, \mathbf{x_r}), \tag{2}$$

where $l \in \{1, \ldots, N\}$, $\mathbf{x_c} = \{x_{ci}\}_{i=1}^{D_1}$ and $\mathbf{x_r} = \{x_{ri}\}_{i=1}^{D_2}$ denote $D_1$ and $D_2$ dimensional inputs from the camera and radar modalities, respectively. Notice that $j$-th output neuron of the perception model becomes $F_j^N(\mathbf{x_c}, \mathbf{x_r})$.

### 4.1.1    Handling Fusion Layer

Beginning at the first fusion layer ($l = 1$), where aggregation of each modality information takes place, the first-order Taylor expansion of $f_j^1$ is applied

$$f_j^1(\mathbf{x_c}, \mathbf{x_r}) = \sum_i \sum_{m \in \{c,r\}} J_{mji}^1 x_{mi} + \underbrace{\left( f_j^1(\tilde{x}_c, \tilde{x}_r) - \sum_i \sum_{m \in \{c,r\}} J_{mji}^1 \tilde{x}_{mi} + \epsilon \right)}_{b_j^1}, \tag{3}$$

where $m$ indexes the modality, $\tilde{x}_c$, $\tilde{x}_r$ represent reference points for camera and radar data, respectively, and $J_{mji}^1$ denotes the Jacobian calculated at the reference points for the modality $m$.

To assess the contribution of each modality to a function output, we try setting the input values of camera and radar data to zero, yielding the following results

$$f_j^1(\mathbf{x_c}, \mathbf{o_r}) = \sum_i J_{cji}^1 x_{ci} + b_j^1, \quad f_j^1(\mathbf{o_c}, \mathbf{x_r}) = \sum_i J_{rji}^1 x_{ri} + b_j^1, \quad f_j^1(\mathbf{o_c}, \mathbf{o_r}) = b_j^1. \tag{4}$$

We let $h_{cj}^l$, $h_{rj}^l$ and $h_{bj}^l$ indicate the modality-specific features from the camera data, radar data and bias at $l$-th layer respectively, i.e., $h_{cj}^1 = \sum_i J_{cji}^1 x_{ci}$, $h_{rj}^1 = \sum_i J_{rji}^1 x_{ri}$, and $h_{bj}^1 = b_j^1$. Then, we can show that

$$f_j^1(\mathbf{x_c}, \mathbf{x_r}) = \sum_{m \in \{c,r,b\}} h_{mj}^1. \tag{5}$$

### 4.1.2    Handling Subsequent Layers

For subsequent layers with $l = 2, \ldots, N$, the $l$-th layer function takes $\sum_{m \in \{c,r,b\}} h_{mj}^{l-1}$ and produces the output

$$f_j^l \left( \sum_{m \in \{c,r,b\}} h_{mj}^{l-1} \right) = \sum_i \sum_{m \in \{c,r,b\}} J_{mji}^l h_{mi}^{l-1} + b_j^l = \sum_{m \in \{c,r,b\}} h_{mj}^l \tag{6}$$

It shows that each layer output results from the summation of the camera, radar, and bias components.

From (6), we can show inductively that the following equation holds

$$F_j^l(\mathbf{x_c}, \mathbf{x_r}) = f_j^l \left( \sum_{m \in \{c,r,b\}} h_{mj}^{l-1} \right) = \sum_{m \in \{c,r,b\}} h_{mj}^l \tag{7}$$

Through modality decomposition, (7) is strictly satisfied across all layers. To effectively decouple the contributions of different modalities, two conditions must be met. First, the decomposition should preserve the functional behavior of the original model. Second, each modality-specific feature must remain unaffected by the others—a condition we refer to as the separation property.

## 4.2   Linearization of Deep Networks

In Section 4.1, we showed that LMD can be derived using a first-order Taylor decomposition. However, when higher-order terms remain, the modality decomposition is difficult to achieve. To circumvent this, we linearize every layer of the fusion model. For the linearized layer functions $\hat{f}^1, \ldots, \hat{f}^N$, our linearization process should preserve the functional behavior of the original network,

$$F_j^l(\mathbf{x_c}, \mathbf{x_r}) = \hat{F}_j^l(\mathbf{x_c}, \mathbf{x_r}), \tag{8}$$

for $l \in 1, \ldots, N$, where $\hat{F}_j^l(\mathbf{x_c}, \mathbf{x_r}) = [\hat{f}^l \circ \cdots \circ \hat{f}^2 \circ \hat{f}^1(\mathbf{x_c}, \mathbf{x_r})]_j$. In other words, our decomposition is designed to satisfy the constraint in (8), ensuring that the linearized network behaves identically to the original one.

In typical neural networks, non-linearities primarily arise from activation and normalization layers. Section 4.2.1 and 4.2.2 describe how these layers are linearized for activation and normalization operations, respectively.

### 4.2.1   Linearizing Activation Layers

We first linearize the activation layers. Specifically, we transform the activation operation into a (linear) element-wise multiplication operation which enables modality decomposition. This process consists of two forward passes.

From the first forward pass, we store the output-to-input ratio $c_j^l$ for $l$-th layer before the linearization. For ReLU [27], this ratio degenerates to a binary mask $c_j^l \in \{0, 1\}$ that records whether $j$-th neuron in the ReLU function $f^l$ was active or not. The main purpose of the first forward pass is to record the activation behavior under the full multimodal input (e.g., camera and radar), which is the target configuration for decomposition.

Then in the second pass, we replace the activation layer $f^l$ with the linearized layer $\hat{f}^l$, which multiplies the input neuron by $c_j^l$. The ultimate goal of the second forward pass is to follow the $f^l$'s behavior even when the input from the specific modality is set to zero to compute the other modality contributions. In other words, the linearization enables the decomposition of the modality features for the target function output which we want to interpret.

**Proposition 1.** *Let $\hat{f}^l$ be the locally linearized activation function,*

$$\hat{f}_j^l(h_{mj}^{l-1}) = c_j^l \cdot h_{mj}^{l-1}, \quad c_j^l := \frac{F_j^l(\mathbf{x_c}, \mathbf{x_r})}{F_j^{l-1}(\mathbf{x_c}, \mathbf{x_r}) + \varepsilon}. \tag{9}$$

*This construction is equivalent to setting the diagonal entries of the $\mathbf{J}_{ji}^l$ in (3) with the slope of the line or segment joining the two operating points $\left(F_j^{l-1}(\mathbf{x_c}, \mathbf{x_r}), F_j^l(\mathbf{x_c}, \mathbf{x_r})\right)$, thereby satisfying (8) exactly.*

The proof can be found in Appendix B.1.

### 4.2.2   Linearizing Normalization Layers

The non-linearity in BatchNorm [28] originates from the bias term. In the case of BatchNorm, statistics stored during the training phase are used as fixed values in the evaluation mode. Thus, aside

from the activation layer, it is not needed to store the behavior of the layer $f^l$ from the forward pass. Consequently, we can linearize the BatchNorm simply by excluding the bias term from the modality features while satisfying (8).

**Proposition 2.** *If $f^l$ is BatchNorm, linearizing $f^l$ yields the following decomposition rule*

$$\hat{f}_j^l\left(h_{mj}^{l-1}\right) = \frac{h_{mj}^{l-1} - \delta_m \mu_j^l}{\sqrt{\sigma_j^{l\,2} + \varepsilon}} \gamma_j^l + \delta_m \beta_j^l = \underbrace{\frac{\gamma_j^l}{\sqrt{\sigma_j^{l\,2} + \varepsilon}} h_{mj}^{l-1}}_{\text{modality specific term}} + \underbrace{\delta_m \left(\beta_j^l - \frac{\mu_j^l \gamma_j^l}{\sqrt{\sigma_j^{l\,2} + \varepsilon}}\right)}_{\text{bias term}} \tag{10}$$

*where $m \in \{c, r, b\}$, $\varepsilon, \mu_j^l, \sigma_j^l, \gamma_j^l, \beta_j^l \in \mathbb{R}$, $\mu_j^l$ and $\sigma_j^l$ denote the batch mean and standard deviation, and $\gamma_j^l, \beta_j^l$ are the affine parameters. In our LMD configuration, $\delta_c$, $\delta_r = 0$ and $\delta_b = 1$, satisfying (8).*

The proof can be found in Appendix B.2.

In contrast to BatchNorm, LayerNorm [29] computes normalization statistics from the current input in both training and evaluation. As noted in [13], LayerNorm can be decomposed into a centering step and a rescaling step. Since the Jacobian in the centering step of LayerNorm is a linear projection matrix $\mathbf{I} - \frac{1}{d}\mathbf{1}\mathbf{1}^\top$, where $\mathbf{I}$ is the identity matrix and $\mathbf{1}$ is an all-one vector, no additional linearization is required.

For the rescaling step, the variance $\sigma^{l\,2} = \text{Var}[F^l(\mathbf{x_c}, \mathbf{x_r})]$ is assumed to be a constant, as the Jacobian entries in rescaling step are near zero, as shown in [13]. Hence, the variance is saved from a single forward pass, and used for the linearized LayerNorm. Even if the interaction effect between modalities remains, modality features maintain their meanings since the rescaling process is applied across all modality features, not altering the importance relationship of modality features. We refer to this approach as the ratio rule, which is adopted in the variants of LMD.

**Proposition 3.** *If $f^l$ is LayerNorm, linearizing $f^l$ yields the following decomposition rule*

$$\hat{f}_j^l\left(h_{mj}^{l-1}\right) = \frac{h_{mj}^{l-1} - \mathbb{E}[h_{mj}^{l-1}]}{\sqrt{\sigma_j^{l\,2} + \varepsilon}} \gamma_j^l + \delta_m \beta_j^l = \underbrace{\frac{\gamma_j^l}{\sqrt{\sigma_j^{l\,2} + \varepsilon}} (h_{mj}^{l-1} - \mathbb{E}[h_{mj}^{l-1}])}_{\text{modality specific term}} + \underbrace{\delta_m \beta_j^l}_{\text{bias term}} \tag{11}$$

*where $m \in \{c, r, b\}$, $\varepsilon, \sigma_j^l, \gamma_j^l, \beta_j^l \in \mathbb{R}$, $\sigma_j^l$ denote the standard deviation, and $\gamma_j^l, \beta_j^l$ are the affine parameters. In our LMD configuration, the expectation is computed over the spatial dimensions (height and width) for each modality and $\delta_c$, $\delta_r = 0$ and $\delta_b = 1$, satisfying (8).*

The proof is given in Appendix B.3. In a similar way, we can also linearize InstanceNorm as shown in Appendix B.3.

### 4.2.3 From Linearization to Modality Decomposition

In this subsection, we formally show that the output of each layer in LMD can be decomposed into modality-specific components. This modality decomposition is possible because all non-linear operations in the network have been replaced with the linear operations as described in Section 4.2.1–4.2.2. As a result, the LMD framework becomes fully linear, allowing the exact separation of modality contributions across all layers in the entire network.

Here, $\hat{f}$ denotes the linearized layer function obtained from the original non-linear function $f$. Eq. (12) thus represents the unified layer formulation after linearization, where activation and normalization operations are already expressed in their linearized forms. Let $h_{cj}^1 = \hat{f}_j^1(\mathbf{x_c}, \mathbf{o_r})$, $h_{rj}^1 = \hat{f}_j^1(\mathbf{o_c}, \mathbf{x_r})$, and $h_{bj}^1 = \hat{f}_j^1(\mathbf{o_c}, \mathbf{o_r})$ be the first layer's modality features. Then, with all the linearized layers in a network, the following equation holds.

$$h_{mj}^l = \hat{f}_j^l(h_{mj}^{l-1}) \tag{12}$$

for all $l \in 2, \dots, N$ and $m \in \{c, r, b\}$.

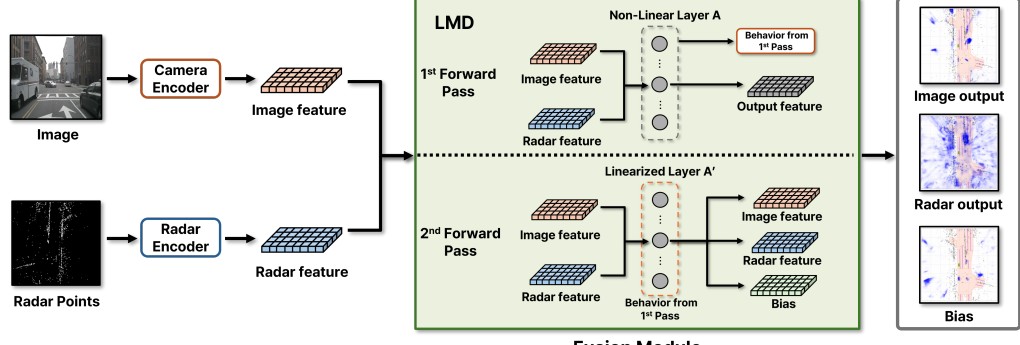

Figure 1: **Overall Process of LMD.** LMD decomposes the multimodal features into modality-specific components through a two-stage process.

Since each layer function $\hat{f}$ becomes linear after applying the linearization process described in Section 4.2.1—4.2.2, the superposition property holds as shown in (13).

$$\hat{f}_j^l \left( \sum_{m \in \{c,r,b\}} h_{mj}^{l-1} \right) = \sum_{m \in \{c,r,b\}} \hat{f}_j^l(h_{mj}^{l-1}) \tag{13}$$

for all $l \in 2, \ldots, N$.

When the entire network is fully linearized as $\hat{F} = \hat{f}^N \circ \cdots \circ \hat{f}^1$, the decomposition extends to the whole network, leading to (14).

$$\hat{F}_j^l(\mathbf{x_c}, \mathbf{x_r}) = \underbrace{\hat{F}_j^l(\mathbf{x_c}, \mathbf{o_r})}_{h_{cj}^l} + \underbrace{\hat{F}_j^l(\mathbf{o_c}, \mathbf{x_r})}_{h_{rj}^l} + \underbrace{\hat{F}_j^l(\mathbf{o_c}, \mathbf{o_r})}_{h_{bj}^l} \tag{14}$$

for all $l \in 1, \ldots, N$.

This guarantees that the LMD framework satisfies both the constraint (8) and the *separation property* throughout the whole linearized network.

### 4.2.4  Exploring LMD Variants

This section presents linearization schemes for normalization layers that satisfy our constraints – Eq.(8) and the *separation property*. For the BatchNorm layer, Proposition 2 excludes the bias term from each modality feature that we call the identity rule. Alternatively, the uniform rule can be applied by uniformly distributing the bias term across all modality features, defined by setting $\delta_m = 1/M$ where $m \in \{c, r, b\}$, and $M = |\{c, r, b\}|$ from (10). To linearize the LayerNorm, in addition to the ratio rule in (11), one can adjust them to operate similarly to BatchNorm from stored statistics calculated from a single forward pass of the original fusion model, where the statistics become $\mu^l = \mathbb{E}[F^l(\mathbf{x_c}, \mathbf{x_r})]$, and $\sigma^{l^2} = \mathrm{Var}[F^l(\mathbf{x_c}, \mathbf{x_r})]$. Then the identity and uniform rules can also be applied. In Section 5.3, we quantitatively evaluate separability across these variants.

### 4.3  Post-hoc Interpretation through LMD

In practice, the LMD is implemented with two forward passes. We store the behaviors of non-linear layer, $c_j^l$ for activation layer and variance in LayerNorm. Then in the second forward pass, we replace the original layer into linearized layer with the cached values. The overall process of LMD is described in Figure 1.

We present the post-hoc interpretation method that uses LMD for the multi-sensor perception model. Specifically, the visualizations in Figure 2 represent $F^N(\mathbf{x_c}, \mathbf{x_r})$, $\hat{F}^N(\mathbf{o_c}, \mathbf{x_r})$, $\hat{F}^N(\mathbf{x_c}, \mathbf{o_r})$, and $\hat{F}^N(\mathbf{o_c}, \mathbf{o_r})$ for the prediction from fused modality and prediction from each modality. Since the prediction from the original model is perfectly decomposed into the prediction from each sensor data,

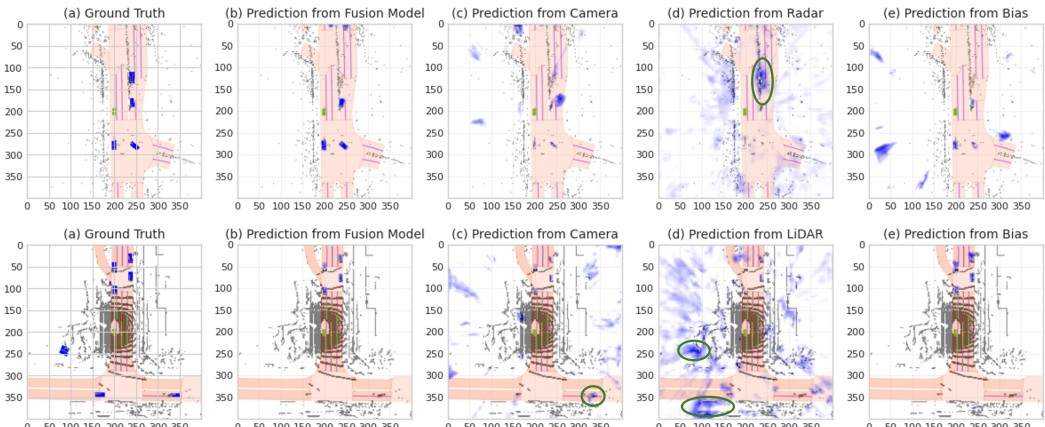

Figure 2: **Post-hoc Interpretation through LMD** : In the first row, a comparison between (c) and (d) shows that the model successfully detects the vehicle using radar data, as indicated by the green marker, whereas the camera-based prediction lacks confidence. Similarly, in the second row, the green marker in (d) highlights either a correct or incorrect prediction made by one modality that was not captured by the other. The prediction from bias in (e) exhibits a certain degree of perceptual capability. This component includes constant effect and high-order interactions largely originated from linearization of activation layers.

Figure 2 shows not just an explanation but an exact description of how pretrained model produces its output exploiting each sensor data. Note that visualizations in Figure 2 extract the positive components of each prediction and then normalize to maximum values to effectively visualize the high-confidence regions. Additional visualizations containing negative values are included in the Appendix D.3.

## 5 Experiments

In this section, we validate the proposed LMD method built on SimpleBEV [23], where modalities are integrated to perform vehicle perception tasks. To demonstrate the generalizability of our approach, we apply LMD to radar-camera, LiDAR-camera and radar-LiDAR-camera fusion models. Details regarding model configurations and the dataset [19] are provided in Appendix A. Also, quantitative results with three-modality settings are provided in Appendix D.1.

Table 1: Comparison of Pearson Correlation and Mean Squared Error for Linearization Experiments.

| Modality | Method | Act. | Norm. | Pearson Correlation | | | | Mean Squared Error | | | |
|---|---|---|---|---|---|---|---|---|---|---|---|
| | | | | $R_p/R$ ($\downarrow$) | $R_p/C$ ($\uparrow$) | $C_p/R$ ($\uparrow$) | $C_p/C$ ($\downarrow$) | $R_p/R$ ($\uparrow$) | $R_p/C$ ($\downarrow$) | $C_p/R$ ($\downarrow$) | $C_p/C$ ($\uparrow$) |
| Radar + Camera | Baseline | ✓ | | $0.22 \pm 0.10$ | $0.76 \pm 0.07$ | $0.22 \pm 0.07$ | $\mathbf{0.09} \pm 0.06$ | $1.46 \pm 0.86$ | $8.51 \pm 19.73$ | $1.46 \pm 1.18$ | $40.05 \pm 23.14$ |
| | | ✓ | | $0.56 \pm 0.09$ | $0.80 \pm 0.07$ | $0.22 \pm 0.07$ | $0.12 \pm 0.07$ | $2.26 \pm 0.82$ | $7.53 \pm 16.54$ | $3.75 \pm 1.25$ | $34.92 \pm 25.52$ |
| | | | ✓ | $0.50 \pm 0.08$ | $0.99 \pm 0.00$ | $0.93 \pm 0.05$ | $0.38 \pm 0.13$ | $8.40 \pm 4.49$ | $0.48 \pm 0.37$ | $0.74 \pm 0.59$ | $41.05 \pm 20.79$ |
| | LMD (Ratio Rule) | ✓ | ✓ | $\mathbf{0.05} \pm 0.02$ | $\mathbf{1.00} \pm 0.00$ | $\mathbf{1.00} \pm 0.00$ | $0.15 \pm 0.04$ | $\mathbf{9.05} \pm 2.34$ | $\mathbf{0.00} \pm 0.00$ | $\mathbf{0.00} \pm 0.00$ | $\mathbf{54.49} \pm 22.15$ |
| LiDAR + Camera | Baseline | ✓ | | $0.41 \pm 0.08$ | $0.77 \pm 0.05$ | $0.31 \pm 0.07$ | $\mathbf{0.11} \pm 0.04$ | $1.46 \pm 10.08$ | $8.51 \pm 10.20$ | $1.46 \pm 8.51$ | $40.05 \pm 11.32$ |
| | | ✓ | | $0.56 \pm 0.08$ | $0.80 \pm 0.05$ | $0.22 \pm 0.07$ | $0.12 \pm 0.05$ | $2.26 \pm 6.77$ | $7.53 \pm 6.83$ | $3.75 \pm 8.81$ | $34.92 \pm 34.17$ |
| | | | ✓ | $0.50 \pm 0.07$ | $0.99 \pm 0.00$ | $0.93 \pm 0.01$ | $0.38 \pm 0.13$ | $8.40 \pm 5.93$ | $0.48 \pm 0.20$ | $0.74 \pm 0.33$ | $\mathbf{41.05} \pm 31.79$ |
| | LMD (Ratio Rule) | ✓ | ✓ | $\mathbf{0.09} \pm 0.03$ | $\mathbf{1.00} \pm 0.00$ | $\mathbf{1.00} \pm 0.00$ | $0.44 \pm 0.06$ | $\mathbf{16.17} \pm 3.39$ | $\mathbf{0.00} \pm 0.00$ | $\mathbf{0.00} \pm 0.00$ | $36.38 \pm 14.52$ |

### 5.1 Metrics to Evaluate the Constraints in LMD

Existing studies rarely attempt modality decomposition in fusion networks, and there is no standard evaluation method to validate such a decomposition. To bridge the gap, we introduce a metric designed to assess how each modality contributes to the model prediction.

We evaluate the model by replacing the input for one modality with an uncorrelated sample while keeping the other modality fixed. Then we compare the linearized predictions before and after

substitution to assess whether the model reflects the change and remains stable for the unchanged modality.

We define two criteria for the assessment: (1) The prediction associated with the perturbed modality should change noticeably, ensuring that the model output reflects the new input rather than being dominated by modality-independent bias. (2) The prediction associated with the unperturbed modality should remain unchanged, indicating that each modality is decoupled from the other modality.

For example, when the radar input is perturbed, the radar-based prediction should present low correlation with its original output, while the camera-based prediction should remain highly correlated with the original prediction. To quantify these effects, we employ the Pearson Correlation Coefficient [30, 31, 32] and Mean Squared Error (MSE) to measure both similarity and distance between modality-specific predictions. Detailed metric formulations and sampling strategy are provided in Appendix D.3.

## 5.2 Evaluating the Linearization Effects in LMD

We evaluate the impact of linearization by comparing the modality-specific predictions before and after the input perturbation. Specifically, $R_p/R$ measures how much the radar-based prediction changes when the radar input is replaced. Since the baseline model lacks decomposition, it requires separate forward passes for clean and perturbed inputs. In contrast, LMD applies a single linearized model derived from clean inputs, ensuring consistent behavior across perturbations. In partial variants, only specific components such as activation layers or normalization parameters are fixed, while the rest of the model remains responsive to input changes. This setup enables a fair comparison, despite architectural differences.

As shown in Table 1, LMD with the ratio rule satisfies both evaluation criteria. When the radar input is replaced, the radar-based output shows a Pearson correlation of 0.05 with its original output and a mean squared error of 9.05, indicating a strong response to the perturbation. Meanwhile, the camera-based output remains unaffected, with a correlation of 1.00. Similar results are observed when perturbing the camera input. These findings demonstrate that LMD effectively isolates the influence of each modality, as each output responds only to changes in its corresponding input. In contrast, baseline models show residual correlation across modalities, implying that the predictions remain entangled and fail to achieve modality separation.

Table 2: Comparison of Pearson Correlation and Mean Squared Error using different LMD variants.

| Modality | BN-LN | Pearson Correlation | | | | Mean Squared Error | | | |
|---|---|---|---|---|---|---|---|---|---|
| | | $R_p/R$ ($\downarrow$) | $R_p/C$ ($\uparrow$) | $C_p/R$ ($\uparrow$) | $C_p/C$ ($\downarrow$) | $R_p/R$ ($\uparrow$) | $R_p/C$ ($\downarrow$) | $C_p/R$ ($\downarrow$) | $C_p/C$ ($\uparrow$) |
| Radar + Camera | Uniform - Identity | $0.50 \pm 0.04$ | $1.00 \pm 0.00$ | $1.00 \pm 0.00$ | $0.42 \pm 0.05$ | $\mathbf{9.58} \pm 2.71$ | $0.00 \pm 0.00$ | $0.00 \pm 0.00$ | $38.89 \pm 10.80$ |
| | Identity - Identity | $0.15 \pm 0.22$ | $1.00 \pm 0.00$ | $1.00 \pm 0.00$ | $0.38 \pm 0.04$ | $\mathbf{9.58} \pm 2.71$ | $0.00 \pm 0.00$ | $0.00 \pm 0.00$ | $38.89 \pm 10.80$ |
| | Identity - Uniform | $0.18 \pm 0.03$ | $1.00 \pm 0.00$ | $0.99 \pm 0.00$ | $0.38 \pm 0.04$ | $9.56 \pm 2.71$ | $0.00 \pm 0.00$ | $0.04 \pm 0.01$ | $38.55 \pm 10.71$ |
| | Identity - Ratio | $\mathbf{0.05} \pm 0.02$ | $\mathbf{1.00} \pm 0.00$ | $\mathbf{1.00} \pm 0.00$ | $\mathbf{0.15} \pm 0.04$ | $9.05 \pm 2.34$ | $\mathbf{0.00} \pm 0.00$ | $\mathbf{0.00} \pm 0.00$ | $54.49 \pm 22.15$ |
| LiDAR + Camera | Uniform - Identity | $0.41 \pm 0.05$ | $1.00 \pm 0.00$ | $1.00 \pm 0.00$ | $0.48 \pm 0.07$ | $15.82 \pm 3.63$ | $0.08 \pm 0.02$ | $0.00 \pm 0.00$ | $31.77 \pm 15.37$ |
| | Identity - Identity | $\mathbf{0.08} \pm 0.04$ | $1.00 \pm 0.00$ | $1.00 \pm 0.00$ | $0.60 \pm 0.07$ | $15.82 \pm 3.63$ | $0.08 \pm 0.02$ | $0.00 \pm 0.00$ | $31.77 \pm 15.37$ |
| | Identity - Uniform | $0.09 \pm 0.04$ | $1.00 \pm 0.00$ | $1.00 \pm 0.00$ | $0.59 \pm 0.07$ | $15.82 \pm 3.61$ | $0.04 \pm 0.01$ | $0.01 \pm 0.00$ | $31.92 \pm 15.39$ |
| | Identity - Ratio | $0.09 \pm 0.03$ | $\mathbf{1.00} \pm 0.00$ | $\mathbf{1.00} \pm 0.00$ | $\mathbf{0.44} \pm 0.06$ | $\mathbf{16.17} \pm 3.39$ | $\mathbf{0.00} \pm 0.00$ | $\mathbf{0.00} \pm 0.00$ | $\mathbf{36.38} \pm 14.52$ |

## 5.3 Quantitative Analysis of *Separation Property* on LMD Variants

Table 2 presents the quantitative results of applying different variants of LMD in both radar-camera and LiDAR-camera fusion models. We evaluate four combinations: *uniform-identity*, *identity-identity*, *identity-uniform*, and *identity-ratio*, while the same metrics applied as in Section 5.2.

Among the tested configurations, the combination of the identity rule for BatchNorm and the ratio rule for LayerNorm exhibits the clearest separation effect. In the radar-camera setting, this configuration yields the lowest correlation for the perturbed modality, with values of 0.05 for $R_p/R$ and 0.15 for $C_p/C$, while maintaining full stability in the unperturbed modality, with $R_p/C$ and $C_p/R$ both equal to 1.00. It also produces zero mean squared error for the fixed modality. Similar behavior is observed in the LiDAR-camera fusion case. Other configurations such as using *uniform-identity* and *identity-uniform* rules result in higher cross-modality correlations and smaller error between clean

and perturbed inputs. This indicates that the outputs are not well separated. These findings suggest that applying the ratio rule is essential for preserving the *separation property* in LMD.

## 6 Discussion

**Computational Efficiency and Scalability** Table 3 summarizes the computational complexity and memory consumption from a single forward pass. LMD requires two forward passes maintaining $\mathcal{O}(1)$ auxiliary state by caching information for linearization, which are discarded immediately after decomposition. LRP-based explanations necessitate an additional backward pass under gradient checkpointing, retain $\mathcal{O}(\sqrt{N_\ell})$ activations across $N_\ell$ layers. Shapley-based approaches preserve low memory but incur exponential computation in the number of modalities, requiring $\mathcal{O}(2^M)$ forward passes to enumerate all coalitions over $M$ modalities.

**LMD + SHAP** As SHAP [33] provides local explanations under an independence assumption among features, stability under cross-modal interactions is examined by the modality-replacement experiment and measuring cross-modality correlations of the resulting predictions, thereby quantifying the consistency of SHAP interpretations in the presence of interactions.

A combination of LMD and SHAP was also evaluated. LMD first decomposes the model prediction into modality-specific components; SHAP is then applied to the bias term, and the resulting attribution is redistributed to the modality-specific predictions. In modality-replacement experiments, this LMD + SHAP better satisfied modality-level separability than SHAP alone. Quantitative results are summarized in Table 4.

Table 3: Computational Complexity and Memory Consumption (Single Forward-pass Measurement). $M$ : number of modalities, $N_L$ : number of layers

| Methods | Passes | Computational Complexity | Memory Consumption |
|---|---|---|---|
| LRP [10] | 2 FWD + 1 BWD | $O(1)$ | $O(\sqrt{N_\ell})$ |
| Shapley-based [33] | $2^M$ FWD | $O(2^M)$ | $O(1)$ |
| LMD | 2 FWD | $O(1)$ | $O(1)$ |

Table 4: Correlation under Modality Replacement: SHAP vs LMD + SHAP.

| Modality | Metric ↑ | SHAP | LMD + SHAP |
|---|---|---|---|
| Radar + Camera | $R_p$/C | $0.6909 \pm 0.11$ | $\mathbf{0.9385} \pm 0.04$ |
| | $C_p$/R | $0.6746 \pm 0.07$ | $\mathbf{0.8942} \pm 0.05$ |
| LiDAR + Camera | $L_p$/C | $0.7062 \pm 0.12$ | $\mathbf{0.9210} \pm 0.04$ |
| | $C_p$/L | $0.7205 \pm 0.06$ | $\mathbf{0.9095} \pm 0.02$ |
| Radar + LiDAR + Camera | $R_pL_p$/C | $0.7108 \pm 0.11$ | $\mathbf{0.8790} \pm 0.05$ |
| | $C_pL_p$/R | $0.7066 \pm 0.08$ | $\mathbf{0.9574} \pm 0.01$ |
| | $C_pR_p$/L | $0.7025 \pm 0.07$ | $\mathbf{0.9389} \pm 0.02$ |

**Applicability to Attention-based Model** LMD can also be applied to an attention-based fusion model. For example, based on the Appendix D.2, we demonstrate LMD on an attention-based radar–camera fusion network, CRN [21] and extract the main term for each modality. Appendix D.2 details how the non-linearities of the attention block's— (bilinear) matrix multiplication, and softmax, can be treated. We also note a limitation: the attention module inevitably introduces bilinear cross–modal terms that cannot be fully assigned to a single modality under LMD framework. Hence, the assessment of the bilinear component requires deeper investigation by future works.

## 7 Conclusions

In this study, we introduced LMD, a framework designed to enhance the interpretability of multimodal fusion models, with a particular focus on multi-sensor perception in autonomous driving context. LMD provides a model-agnostic approach that clearly attributes model predictions to individual sensor modalities. By disentangling modality-specific contributions across network layers, LMD offers deeper insights into how each modality influences the fusion process. Our qualitative and quantitative evaluations demonstrated the effectiveness of LMD in identifying modality-dependent behaviors, revealing when and where fusion improves—or potentially degrades—model performance. Overall, this work underscored the critical role of interpretability in autonomous driving systems, contributing to greater transparency in model decision-making and laying the groundwork for future research on interpretable multimodal learning.

## Acknowledgments

This work was supported by 1) Institute of Information & communications Technology Planning & Evaluation (IITP) grant funded by the Korea government (MSIT) [NO.RS-2021-II211343, Artificial Intelligence Graduate School Program (Seoul National University)] and 2) the Ministry of Trade, Industry and Energy (MOTIE, Korea) (No.RS-2024-00443216, Development and PoC of On-Device AI Computing based AI Fusion Mobility Device).

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

# Appendix

## A Implementation Details

This section introduces the details utilized to implement LMD. First, we describe the dataset used for the BEV perception model employed to demonstrate our method, as well as the configuration of the fusion model utilized in the experiments.

### A.1 Dataset

The nuScenes dataset [19] is a large-scale, multimodal dataset designed for autonomous driving tasks. It encompasses 1000 scenes, segmented into 700 for training, 150 for validation, and 150 for testing, each lasting about 20 seconds. The dataset captures a 360° horizontal field of view (FOV) through the use of six surround-view cameras, one lidar, and five radars, providing comprehensive environmental perception.

Notably, the nuScenes dataset stands out due to its high annotation frequency of every 0.5 seconds, translating to a 2 Hz rate, which includes over 1.4 million 3D bounding boxes across ten object categories. This extensive annotation facilitates advanced tasks like 3D object detection and tracking.

For evaluating detection performance, the dataset introduces unique metrics, including mean Average Precision (mAP) and five true positive (TP) metrics (ATE, ASE, AOE, AVE, and AAE) to measure errors in translation, scale, orientation, velocity, and attribute, respectively. These metrics are aggregated into the nuScenes Detection Score (NDS) to provide a comprehensive performance overview.

Furthermore, for the task like the Bird's Eye View (BEV) segmentation, the dataset follows specific settings proposed in prior research, leveraging its rich radar point cloud data among other modalities. The nuScenes dataset not only facilitates the development of advanced autonomous driving technologies but also sets a benchmark for evaluating the performance of these systems through its detailed and nuanced evaluation metrics.

### A.2 Model Architecture & Hyper-Parameters

The basic settings of the camera-radar fusion model follow those of SimpleBEV [23]. The camera branch processes input from six view images, each of which is passed through a backbone network, such as ResNet [34], to extract image features. These image features, along with radar point cloud information, are subsequently mapped onto the BEV coordinate system to facilitate perception tasks in the BEV space. The camera features and radar features in the BEV representation are concatenated and processed using convolutional operations to generate latent features. In this paper, this convolution process is defined as the fusion operation performed by the layer function ($l = 1$). The latent features are then passed through a U-Net [35] decoder, after which the prediction head carries out a segmentation task focused on identifying vehicle regions. table 5 presents radar-camera and LiDAR-camera configuration details used for the SimpleBEV framework.

Table 5: Detailed configuration of modality-specific fusion setups

| Parameter | Radar / LiDAR - Camera |
|---|---|
| Backbone | ResNet101 |
| Fusion Operation | Concat & Conv |
| Sweeps | 5 |
| Input Size | (224, 400) |
| BEV Coordinate | (200, 8, 200) |

### A.3 Experiments Compute Resources

We employed pre-trained encoders for camera, radar, and LiDAR modalities. Our method was evaluated for 6019 iterations with a batch size of 1 on 4 x NVIDIA RTX 3090 GPUs, which required approximately 3 hours in total.

# B   Proofs on LMD

## B.1   Proposition 1: Handling Activation Layers

Let $A_j = F_j^{l-1}(x_c, x_r)$ denote the total input to the activation function for neuron $j$ in layer $l$. Let $O_j = F_j^l(x_c, x_r)$ denote the original output of this activation function for neuron $j$ in layer $l$. The linearized activation function for a modality-specific component $h_{mj}^{l-1}$ is defined as $\hat{f}_j^l(h_{mj}^{l-1}) = c_j \cdot h_{mj}^{l-1}$. The constant $c_j$ is given by $c_j = \frac{O_j}{A_j + \varepsilon}$.

The total input to the activation function is the sum of its decomposed modality-specific components: $A_j = \sum_{m \in \{c,r,b\}} h_{mj}^{l-1}$. The output of the linearized layer $\hat{F}_j^l(x_c, x_r)$ is obtained by summing the outputs of the linearized activation function applied to each component (due to the linearity of $h \mapsto c_j h$ and as implied by eq. (12) and eq. (13) which state $h_{mj}^l = \hat{f}_j^l(h_{mj}^{l-1})$ and $\hat{f}_j^l(\sum_{m \in \{c,r,b\}} h_{mj}^{l-1}) = \sum_{m \in \{c,r,b\}} \hat{f}_j^l(h_{mj}^{l-1})$):

$$\hat{F}_j^l(x_c, x_r) = \sum_{m \in \{c,r,b\}} \hat{f}_j^l(h_{mj}^{l-1}) = \sum_{m \in \{c,r,b\}} c_j h_{mj}^{l-1} = c_j \sum_{m \in \{c,r,b\}} h_{mj}^{l-1} = c_j A_j.$$

Substituting the definition of $c_j$:

$$\hat{F}_j^l(x_c, x_r) = \left(\frac{O_j}{A_j + \varepsilon}\right) A_j.$$

eq. (8) requires that the linearized model's output equals the original model's output at the operating point: $\hat{F}_j^l(x_c, x_r) = F_j^l(x_c, x_r)$, which is $\hat{F}_j^l(x_c, x_r) = O_j$. Therefore, we must have:

$$O_j = \left(\frac{O_j}{A_j + \varepsilon}\right) A_j.$$

We analyze this equation in different cases:

- **Case 1: $O_j \neq 0$.** In this case, we can divide both sides by $O_j$:

$$1 = \frac{A_j}{A_j + \varepsilon}.$$

  This implies $A_j + \epsilon = A_j$, which means $\epsilon = 0$. Thus, if $O_j \neq 0$, eq. (8) is satisfied when $\epsilon = 0$.

- **Case 2: $O_j = 0$.** The equation becomes:

$$0 = \left(\frac{0}{A_j + \varepsilon}\right) A_j.$$

  This simplifies to $0 = 0$, provided that $A_j + \varepsilon \neq 0$. This condition typically holds as $\varepsilon$ is a small constant used to prevent division by zero. This case covers scenarios such as:

  - $A_j = 0$ and $O_j = 0$ (e.g., ReLU(0)=0, GELU(0)=0). Here $c_j = 0/(0 + \varepsilon) = 0$. Then $\hat{F}_j^l(x_c, x_r) = 0 \cdot 0 = 0$, which equals $O_j$. eq. (8) holds.
  - $A_j \neq 0$ but $O_j = 0$ (e.g., ReLU applied to a negative input). Here $c_j = 0/(A_j + \varepsilon) = 0$ (assuming $A_j + \varepsilon \neq 0$). Then $\hat{F}_j^l(x_c, x_r) = 0 \cdot A_j = 0$, which equals $O_j$. eq. (8) holds.

The connection to DTD, where $c_j$ is interpreted as the slope of the segment joining $(A_j, O_j)$ (and possibly the origin, if the activation passes through it), underpins this choice. If $c_j = O_j/A_j$ (for $A_j \neq 0$), then $\hat{F}_j^l = (O_j/A_j)A_j = O_j$, directly satisfying eq. (8). The formulation with $\epsilon$ makes this robust.

Thus, the proposed construction for the linearized activation function $\hat{f}_j^l$ ensures that eq. (8) is satisfied under the conditions discussed. □

## B.2   Proposition 2: Handling BatchNorm Layers

The linearized output of the layer is the sum over modality components:

$$\hat{F}_j^l(x_c, x_r) = \sum_{m \in \{c,r,b\}} \hat{f}_j^l(h_{mj}^{l-1}).$$

**Camera ($m = c$) and radar ($m = r$) components.** Because $\delta_c = \delta_r = 0$,

$$\hat{f}_j^l(h_{mj}^{l-1}) = \frac{\gamma_j^l}{\sqrt{(\sigma_j^l)^2 + \varepsilon}} h_{mj}^{l-1} \qquad (m \in \{c, r\}).$$

**Bias component** ($m = b$). With $\delta_b = 1$,

$$\hat{f}_j^l(h_{bj}^{l-1}) = \frac{\gamma_j^l}{\sqrt{(\sigma_j^l)^2 + \varepsilon}}(h_{bj}^{l-1} - \mu_j^l) + \beta_j^l.$$

Summing the three parts yields

$$\hat{F}_j^l(x_c, x_r) = \frac{\gamma_j^l}{\sqrt{(\sigma_j^l)^2 + \varepsilon}}\left(h_{cj}^{l-1} + h_{rj}^{l-1} + h_{bj}^{l-1} - \mu_j^l\right) + \beta_j^l.$$

Because the LMD framework guarantees $F_j^{l-1}(x_c, x_r) = \sum_{m \in \{c,r,b\}} h_{mj}^{l-1}$, we can rewrite the above as

$$\hat{F}_j^l(x_c, x_r) = \frac{\gamma_j^l}{\sqrt{(\sigma_j^l)^2 + \varepsilon}}\left(F_j^{l-1}(x_c, x_r) - \mu_j^l\right) + \beta_j^l,$$

which is the BatchNorm forward pass $F_j^l(x_c, x_r)$ in evaluation mode. Hence $\hat{F}_j^l = F_j^l$, satisfying eq. (8).

## B.3  Proposition 3: Handling InstanceNorm Layers

Let $F^{l-1}(x_c, x_r)$ denote the input of the feature map to the InstanceNorm layer in layer $l$. For a specific channel $j$ within this feature map, let $H_j^{l-1}$ represent the activations for the channel $j$. The standard InstanceNorm operation for this channel is defined as:

$$F_j^l = \gamma_j^l \frac{H_j^{l-1} - \mathbb{E}[H_j^{l-1}]}{\sqrt{\mathrm{Var}[H_j^{l-1}] + \varepsilon}} + \beta_j^l$$

where $\mathbb{E}[H_j^{l-1}]$ and $\mathrm{Var}[H_j^{l-1}]$ are the mean and variance of $H_j^{l-1}$ computed over its spatial dimensions, respectively. $\gamma_j$ and $\beta_j$ are learnable scaling and shifting parameters for channel $j$.

In the LMD framework, the input to the $l$-th layer, $F^{l-1}(x_c, x_r)$, is decomposed into modality-specific components:

$$F^{l-1}(x_c, x_r) = \sum_{m \in \{c,r,b\}} h_m^{l-1}$$

where $h_m^{l-1}$ represents the feature map attributed to the modality $m$ (camera, radar, or bias) at layer $l-1$. For channel $j$, this means:

$$H_j^{l-1} = \sum_{m \in \{c,r,b\}} h_{mj}^{l-1}$$

The LMD framework requires that the linearized layer function $\hat{f}^l$ satisfies the property that the sum of the decomposed outputs equals the output of the linearized function applied to the sum of inputs, which in turn must match the original function output at the operating point from eq. (8).

For InstanceNorm, we state that the variance $\sigma_j^l = \mathrm{Var}[H_j^{l-1}]$ is treated as a constant, saved from a single forward pass of the original fusion model.

By considering the proposed decomposition rule for $\hat{f}_j^l(h_{mj}^{l-1})$:

$$\hat{f}_j^l(h_{mj}^{l-1}) = \frac{h_{mj}^{l-1} - \mathbb{E}[h_{mj}^{l-1}]}{\sqrt{\sigma_j^{l^2} + \varepsilon}}\gamma_j^l + \delta_m \beta_j^l$$

To sum these decomposed outputs over all modalities $m \in \{c, r, b\}$ to see if they reconstruct the original InstanceNorm output $F_j^l$:

$$\sum_{m \in \{c,r,b\}} \hat{f}_j^l(h_{mj}^{l-1}) = \sum_{m \in \{c,r,b\}} \left( \frac{h_{mj}^{l-1} - \mathbb{E}[h_{mj}^{l-1}]}{\sqrt{\sigma_j^{l^2} + \varepsilon}}\gamma_j^l + \delta_m \beta_j^l \right)$$

By splitting the summation:

$$= \frac{\gamma_j^l}{\sqrt{{\sigma_j^l}^2 + \varepsilon}} \sum_{m \in \{c,r,b\}} (h_{mj}^{l-1} - \mathbb{E}[h_{mj}^{l-1}]) + \sum_{m \in \{c,r,b\}} \delta_m \beta_j^l$$

For the first term, we use the linearity of expectation:

$$\sum_{m \in \{c,r,b\}} (h_{mj}^{l-1} - \mathbb{E}[h_{mj}^{l-1}]) = \sum_{m \in \{c,r,b\}} h_{mj}^{l-1} - \sum_{m \in \{c,r,b\}} \mathbb{E}[h_{mj}^{l-1}]$$

$$= H_j^{l-1} - \mathbb{E}\left[ \sum_{m \in \{c,r,b\}} h_{mj}^{l-1} \right]$$

$$= H_j^{l-1} - \mathbb{E}[H_j^{l-1}]$$

For the second term, within the definition of $\delta_m$: $\delta_m = 0$ for $m \in \{c,r\}$ and $\delta_m = 1$ for $m \in \{b\}$. Therefore:

$$\sum_{m \in \{c,r,b\}} \delta_m \beta_j^l = \delta_c \beta_j^l + \delta_r \beta_j^l + \delta_b \beta_j^l = 0 \cdot \beta_j^l + 0 \cdot \beta_j^l + 1 \cdot \beta_j^l = \beta_j^l$$

Substituting these back into the summed expression:

$$\sum_{m \in \{c,r,b\}} \hat{f}_j^l(h_{mj}^{l-1}) = \frac{\gamma_j^l (H_j^{l-1} - \mathbb{E}[H_j^{l-1}])}{\sqrt{{\sigma_j^l}^2 + \varepsilon}} + \beta_j^l$$

Since $\sigma_j^l$ is taken as the fixed variance $\text{Var}[H_j^{l-1}]$ from the first forward pass, this expression is the original InstanceNorm output $F_j^l$. Since each modality is processed independently except for the shared, fixed scale, the resulting features adhere to the intended *separation property*. The proposed decomposition rule for InstanceNorm layers ensures that the sum of the modality-specific outputs from the linearized layer perfectly reconstructs the output of the original InstanceNorm layer, satisfying eq. (8) of the LMD framework.

## C   Variants on Activation Layers

Necessities of considering the bias-splitting rules in activation layers, aside from cases where a constant term is inevitably introduced (e.g., normalization layers), is as follows. When a model is linearized, a bias' prediction can be obtained by inputting zeros into all data points. As shown in Figure 2, it is evident that bias features also exhibit perception capabilities. Therefore, the process of appropriately distributing the bias contribution between the camera and radar can be considered. Particularly in the activation layers, if a specific neuron is activated by the camera feature at that neuron could enhance the camera's contribution. Conversely, if a neuron is activated by the radar, adding a positive bias to the radar feature could strengthen the radar's contribution. This idea led to the development of the sum rule introduced in appendix C.1. The modality predictions based on various bias-splitting rules need to be extensively explored in future experiments.

While the sum rule adds the entire bias to whichever modality actually triggers a ReLU, it disregards how strongly the two modalities contribute in magnitude. Empirically this may still blur modality separation when both camera and radar deliver noticeable—but differently scaled—signals. The ratio splitting rule therefore apportions the bias term $h_{bj}^{l-1}$ proportionally to the absolute pre-activations of camera and radar. The impact of sum and ratio splitting rules can be assessed via the four perturbation-based metrics ($R_p/R$, $R_p/C$, $C_p/R$, $C_p/C$). These rules modulate how sensitively and independently each modality responds to perturbation, offering an empirical basis for bias handling in activation layers.

### C.1   Sum Splitting Rule

For the most common activation function, ReLU, if a camera feature value is less than zero but a radar feature value is greater than zero, resulting in the activation of a specific neuron, it is reasonable to interpret that the neuron was activated by the radar. In such cases, a splitting rule can be applied where the bias feature is added to the radar feature when the bias feature value is greater than zero. Conversely, if both a camera feature value and a bias feature value are greater than zero while the radar feature value is less than zero, distributing the bias feature to the camera feature strengthen the camera's contribution to that neuron. The sum condition tested in our experiments are derived from this idea. The related equations are as follows:

$$Cam - Condition = h_{cj}^{l-1} < 0, h_{rj}^{l-1} > 0, h_{bj}^{l-1} < 0$$
$$\text{and } h_{cj}^{l-1} > 0, h_{rj}^{l-1} < 0, h_{bj}^{l-1} > 0$$

$$Rad - Condition = h_{cj}^{l-1} < 0, h_{rj}^{l-1} > 0, h_{bj}^{l-1} > 0$$
$$\text{and } h_{cj}^{l-1} > 0, h_{rj}^{l-1} < 0, h_{bj}^{l-1} < 0$$

$$\hat{f}^l(h_{cj}^{l-1}) = \begin{cases} c\left(h_{cj}^{l-1} + h_{bj}^{l-1}\right), & \text{for } Cam - Condition, \\ c\left(h_{cj}^{l-1}\right), & \text{otherwise.} \end{cases}$$

$$\hat{f}^l(h_{rj}^{l-1}) = \begin{cases} c\left(h_{rj}^{l-1} + h_{bj}^{l-1}\right), & \text{for } Rad - Condition, \\ c\left(h_{rj}^{l-1}\right), & \text{otherwise.} \end{cases} \tag{15}$$

$$\hat{f}^l(h_{bj}^{l-1}) = \begin{cases} 0, & \text{for } Cam \,\&\, Rad - Condition, \\ c\left(h_{bj}^{l-1}\right), & \text{otherwise.} \end{cases}$$

$$\text{, where} \quad c = \frac{F_j^l(\mathbf{x_c}, \mathbf{x_r})}{F_j^{l-1}(\mathbf{x_c}, \mathbf{x_r}) + \varepsilon}.$$

## C.2 Ratio Splitting Rule

In activation layers a bias term $h_{bj}^{l-1}$ can enhance the modality responsible for the neuron's activation. The ratio splitting rule redistributes that bias to the camera and radar features in proportion to their absolute magnitudes, while leaving neurons that fail the ratio conditions unchanged. This preserves the conservation property in eq. (8) and generalizes the identity and uniform rules introduced earlier.

$$Ratio - Condition = \left(h_{cj}^{l-1} > 0, h_{rj}^{l-1} > 0, h_{bj}^{l-1} > 0\right)$$
$$\text{or } \left(h_{cj}^{l-1} < 0, h_{rj}^{l-1} < 0, h_{bj}^{l-1} < 0\right),$$

$$Ratio - Condition2 = \left(h_{cj}^{l-1} > 0, h_{rj}^{l-1} > 0, h_{bj}^{l-1} < 0\right)$$
$$\text{or } \left(h_{cj}^{l-1} < 0, h_{rj}^{l-1} < 0, h_{bj}^{l-1} > 0\right).$$

$$\alpha_j = \frac{|h_{rj}^{l-1}|}{|h_{cj}^{l-1}| + |h_{rj}^{l-1}| + \varepsilon}, \qquad \alpha_j \in [0, 1],$$

with a small $\varepsilon > 0$ for numerical stability.

Let $c = \dfrac{F_j^l(x_c, x_r)}{F_j^{l-1}(x_c, x_r) + \varepsilon}$ be the slope from proposition 1. Then

$$\hat{f}^l(h_{cj}^{l-1}) = \begin{cases} c\left(h_{cj}^{l-1} + (1 - \alpha_j)\, h_{bj}^{l-1}\right), & Ratio - Condition, \\ c\left(h_{cj}^{l-1} + \alpha_j\, h_{bj}^{l-1}\right), & Ratio - Condition2, \\ c\, h_{cj}^{l-1}, & \text{otherwise,} \end{cases}$$

$$\hat{f}^l(h_{rj}^{l-1}) = \begin{cases} c\left(h_{rj}^{l-1} + \alpha_j\, h_{bj}^{l-1}\right), & Ratio - Condition, \\ c\left(h_{rj}^{l-1} + (1 - \alpha_j)\, h_{bj}^{l-1}\right), & Ratio - Condition2, \\ c\, h_{rj}^{l-1}, & \text{otherwise,} \end{cases}$$

$$\hat{f}^l(h_{bj}^{l-1}) = \begin{cases} 0, & Ratio - Condition \text{ or } Ratio - Condition2, \\ c\, h_{bj}^{l-1}, & \text{otherwise.} \end{cases}$$

Because $h_{bj}^{l-1}$ is only re-partitioned, the eq. (8) is strictly preserved.

The ratio rule thus interpolates smoothly between existing bias-handling strategies while adapting to the magnitude of each modality's evidence, leading to the stability improvements reported in section 5.

Table 6: Comparison of Pearson Correlation and Mean Squared Error for selected LMD rules (Radar + Camera)

| Modality | Method | Pearson Correlation | | | | Mean Squared Error | | | |
|---|---|---|---|---|---|---|---|---|---|
| | | $R_p$/R ($\downarrow$) | $R_p$/C ($\uparrow$) | $C_p$/R ($\uparrow$) | $C_p$/C ($\downarrow$) | $R_p$/R ($\uparrow$) | $R_p$/C ($\downarrow$) | $C_p$/R ($\downarrow$) | $C_p$/C ($\uparrow$) |
| | **Activation** | | | | | | | | |
| Radar + Camera | Sum-Splitting | $0.12 \pm 0.04$ | $1.00 \pm 0.00$ | $1.00 \pm 0.00$ | $0.23 \pm 0.05$ | $\mathbf{9.63} \pm \mathbf{2.34}$ | $0.00 \pm 0.00$ | $0.00 \pm 0.00$ | $\mathbf{40.43} \pm \mathbf{19.35}$ |
| | Ratio-Splitting | $\mathbf{0.08} \pm \mathbf{0.06}$ | $\mathbf{1.00} \pm \mathbf{0.00}$ | $\mathbf{1.00} \pm \mathbf{0.00}$ | $\mathbf{0.18} \pm \mathbf{0.04}$ | $4.46 \pm 1.45$ | $\mathbf{0.00} \pm \mathbf{0.00}$ | $\mathbf{0.00} \pm \mathbf{0.00}$ | $30.35 \pm 4.19$ |

# D Further Experiments

## D.1 Three Modality Experiments

We extend LMD to a three-sensor setting (camera + radar + LiDAR) and performed a comprehensive variant study on bias-splitting strategies. This experiment is expected to further demonstrate the generality of LMD. For brevity, we share the correlation results in tables 7 and 8 using the same metrics as in the main paper. Consistent with the main text, we verify that perturbing one modality does not affect the other. You can refer to the relevant theoretical formulation in Appendix F.

Table 7: Three-modality (R+L+C) ablation of LMD versus baselines

| Method | Act. | Norm. | $C_pR_p$/L ($\uparrow$) | $C_pR_p$/C ($\downarrow$) | $C_pR_p$/R ($\downarrow$) | $C_pL_p$/R ($\uparrow$) | $C_pL_p$/C ($\downarrow$) | $C_pL_p$/L ($\downarrow$) | $R_pL_p$/C ($\uparrow$) | $R_pL_p$/R ($\downarrow$) | $R_pL_p$/L ($\downarrow$) |
|---|---|---|---|---|---|---|---|---|---|---|---|
| Baseline 1 | | | $0.3606 \pm 0.095$ | $0.0679 \pm 0.069$ | $0.1103 \pm 0.095$ | $0.1957 \pm 0.092$ | $0.0955 \pm 0.069$ | $0.2265 \pm 0.094$ | $0.6794 \pm 0.099$ | $0.4755 \pm 0.092$ | $0.4663 \pm 0.104$ |
| Baseline 2 | $\checkmark$ | | $0.9764 \pm 0.018$ | $0.5459 \pm 0.118$ | $0.6565 \pm 0.134$ | $0.9717 \pm 0.029$ | $0.5488 \pm 0.117$ | $0.5360 \pm 0.103$ | $0.9865 \pm 0.009$ | $0.6722 \pm 0.107$ | $0.5384 \pm 0.092$ |
| Baseline 3 | | $\checkmark$ | $0.3685 \pm 0.089$ | $0.0669 \pm 0.072$ | $0.1245 \pm 0.086$ | $0.2165 \pm 0.084$ | $0.2165 \pm 0.084$ | $0.0570 \pm 0.073$ | $0.6880 \pm 0.098$ | $0.4751 \pm 0.092$ | $0.4678 \pm 0.104$ |
| **LMD (Ratio Rule)** | $\checkmark$ | $\checkmark$ | $\mathbf{1.0000} \pm \mathbf{0.000}$ | $\mathbf{0.1385} \pm \mathbf{0.232}$ | $\mathbf{0.0279} \pm \mathbf{0.042}$ | $\mathbf{1.0000} \pm \mathbf{0.000}$ | $\mathbf{0.1488} \pm \mathbf{0.132}$ | $\mathbf{0.2310} \pm \mathbf{0.057}$ | $\mathbf{1.0000} \pm \mathbf{0.000}$ | $\mathbf{0.0373} \pm \mathbf{0.091}$ | $\mathbf{0.2023} \pm \mathbf{0.097}$ |

Table 8: Three-modality (R+L+C) ablation of LMD variants.

| Method | $C_pR_p$/L ($\uparrow$) | $C_pR_p$/C ($\downarrow$) | $C_pR_p$/R ($\downarrow$) | $C_pL_p$/R ($\uparrow$) | $C_pL_p$/C ($\downarrow$) | $C_pL_p$/L ($\downarrow$) | $R_pL_p$/C ($\uparrow$) | $R_pL_p$/R ($\downarrow$) | $R_pL_p$/L ($\downarrow$) |
|---|---|---|---|---|---|---|---|---|---|
| Uniform - Identity | $\mathbf{1.0000} \pm \mathbf{0.000}$ | $0.3224 \pm 0.148$ | $0.5572 \pm 0.114$ | $\mathbf{1.0000} \pm \mathbf{0.000}$ | $0.3164 \pm 0.168$ | $0.3162 \pm 0.043$ | $\mathbf{1.0000} \pm \mathbf{0.000}$ | $0.5572 \pm 0.114$ | $0.3322 \pm 0.089$ |
| Identity - Identity | $\mathbf{1.0000} \pm \mathbf{0.000}$ | $0.3315 \pm 0.170$ | $0.0971 \pm 0.056$ | $\mathbf{1.0000} \pm \mathbf{0.000}$ | $0.3305 \pm 0.161$ | $0.0039 \pm 0.139$ | $\mathbf{1.0000} \pm \mathbf{0.000}$ | $0.0926 \pm 0.080$ | $0.0042 \pm 0.113$ |
| Identity - Uniform | $0.9994 \pm 0.008$ | $0.3332 \pm 0.160$ | $0.0807 \pm 0.008$ | $0.9903 \pm 0.079$ | $0.3182 \pm 0.145$ | $0.0087 \pm 0.1125$ | $0.9999 \pm 0.000$ | $0.0870 \pm 0.078$ | $0.0085 \pm 0.113$ |
| Identity - Ratio | $\mathbf{1.0000} \pm \mathbf{0.000}$ | $\mathbf{0.1385} \pm \mathbf{0.232}$ | $\mathbf{0.0279} \pm \mathbf{0.042}$ | $\mathbf{1.0000} \pm \mathbf{0.000}$ | $\mathbf{0.1488} \pm \mathbf{0.132}$ | $\mathbf{0.2310} \pm \mathbf{0.057}$ | $\mathbf{1.0000} \pm \mathbf{0.000}$ | $\mathbf{0.0373} \pm \mathbf{0.091}$ | $\mathbf{0.2023} \pm \mathbf{0.097}$ |

## D.2 Application to Attention-based Model

Here, we present the experimental results and limitations of applying LMD to attention-based model [21]. In attention operation, two major sources of nonlinearity must be considered : the (bilinear) matrix multiplication and the softmax. For matrix multiplications, we treat a term as a modality feature only when it multiplies same-modality features or a modality feature with a constant; otherwise, it is assigned to the bias features. For the softmax function, we use the same procedure as for linearizing activation layers.

We successfully decomposed modality-specific terms in CRN and observed that high-order term remains due to matrix multiplication between input variables.

Table 9: Comparison (Pearson Correlation) of LMD Variants with CRN [21]

| Modality | Method | $R_p$/R ($\downarrow$) | $R_p$/C ($\uparrow$) | $C_p$/R ($\uparrow$) | $C_p$/C ($\downarrow$) |
|---|---|---|---|---|---|
| Camera + Radar | Uniform – Identity | $0.6123 \pm 0.043$ | $1.0000 \pm 0.000$ | $1.0000 \pm 0.000$ | $0.5912 \pm 0.052$ |
| | Identity – Identity | $0.2184 \pm 0.125$ | $1.0000 \pm 0.000$ | $1.0000 \pm 0.000$ | $0.3891 \pm 0.043$ |
| | Identity – Uniform | $0.2572 \pm 0.067$ | $1.0000 \pm 0.000$ | $0.9882 \pm 0.010$ | $0.3827 \pm 0.046$ |
| | Identity – Ratio | $0.0552 \pm 0.0213$ | $1.0000 \pm 0.000$ | $1.0000 \pm 0.000$ | $0.1571 \pm 0.045$ |

Table 10: Comparison (Mean Squared Error) of LMD Variants with CRN [21]

| Modality | Method | $R_p$/R ($\uparrow$) | $R_p$/C ($\downarrow$) | $C_p$/R ($\downarrow$) | $C_p$/C ($\uparrow$) |
|---|---|---|---|---|---|
| Camera + Radar | Uniform – Identity | $8.9732 \pm 2.142$ | $0.0000 \pm 0.000$ | $0.0000 \pm 0.000$ | $39.4135 \pm 19.802$ |
| | Identity – Identity | $9.4676 \pm 2.295$ | $0.0000 \pm 0.000$ | $0.0000 \pm 0.000$ | $28.8149 \pm 18.801$ |
| | Identity – Uniform | $10.5438 \pm 1.714$ | $0.0000 \pm 0.000$ | $0.0087 \pm 0.041$ | $27.5582 \pm 16.711$ |
| | Identity – Ratio | $11.5812 \pm 2.712$ | $0.0000 \pm 0.000$ | $0.0000 \pm 0.000$ | $54.4890 \pm 22.151$ |

## D.3 Further Post-hoc Interpretation

This section presents additional examples of applying LMD to the BEV perception task. Through further visualizations, we expect that our proposed method is validated with sufficient consistency to reliably interpret the model.

fig. 3 and fig. 4 show further visualization results for the reliability of the proposed LMD on the BEV perception benchmark. fig. 3 normalizes each modality-specific feature map with a sigmoid transform before projection, so only positive activations remain; the bright regions therefore mark locations where a single sensor delivers decisive, supportive cues to the detector, making the saliency of true objects immediately apparent. By contrast, fig. 4 retains the full signed output and colour-codes positive and negative responses, allowing us to see not only where a modality reinforces the final prediction but also where it actively suppresses prediction.

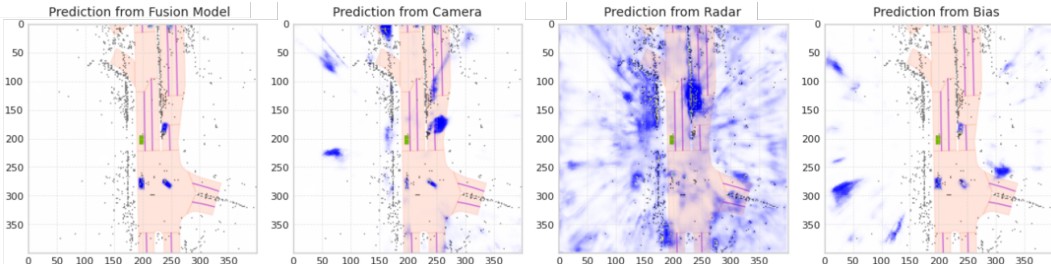

Figure 3: Visualizations of using Sigmoid.

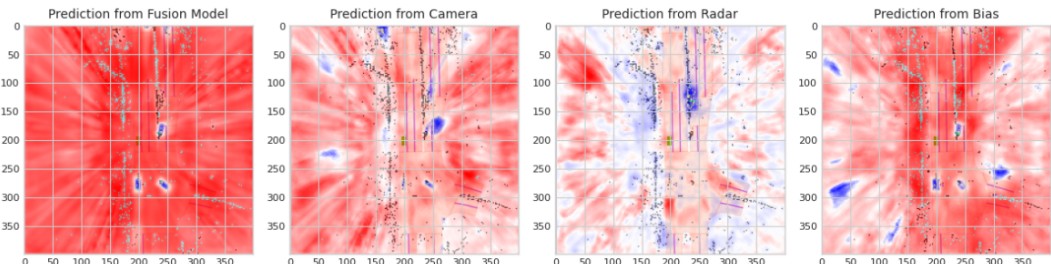

Figure 4: Visualizations of both Positive and Negative Values.

**Interpreting the Perceptual Capability of the Bias Component**   In our setting, the prediction from bias is generated solely from the internal constants without any modality input. Its perceptual capability mainly stems from the linearized activation layers: internal constant components follow the pre-determined activation patterns, yielding structured responses. On the other hand, linearized normalization layers are responsible only for scaling of features and thus they are unlikely to contribute to perceptual expressivity of the bias term.

### D.4   Metric Formulas

To evaluate the separation property of the modality-specific decomposition, we introduce four perturbation-based metrics. Here, perturbation refers to the replacement of one modality input with an uncorrelated sample. These metrics assess how much each modality-specific output responds to changes in its own input and remains invariant to others. Each metric is computed using the Pearson Correlation Coefficient (PCC) and Mean Squared Error (MSE).

**Pearson Correlation Coefficient (PCC).**   Briefly, for pearson correlation coefficient, we define:

$$\text{PCC}_{k,m} = \frac{\sum_i \left(P_k[i] - \bar{P}_k\right)\left(P_{(k+500m)\bmod N}[i] - \bar{P}_{(k+500m)\bmod N}\right)}{\sqrt{\sum_i \left(P_k[i] - \bar{P}_k\right)^2 \sum_i \left(P_{(k+500m)\bmod N}[i] - \bar{P}_{(k+500m)\bmod N}\right)^2}}, \tag{16}$$

where $N = 6019$, $P_k$ and $P_{(k+500m)\bmod N}$ are the predictions for the sample spaced by $500m$ from $k$, $\bar{P}_k$ and $\bar{P}_{(k+500m)\bmod N}$ are their respective mean values, and $m \in \{1, 2, \ldots, 12\}$ ensures 12 distinct comparisons. PCC measures the similarity between two prediction vectors by first subtracting their respective means (mean-centering), computing the dot product of the centered vectors, and normalizing it by the product of their standard deviations. The resulting value lies in the range $[-1, 1]$, where 1 indicates perfect correlation, 0 indicates no correlation, and $-1$ indicates perfect inverse correlation.

**Mean Squared Error (MSE).**   In addition to PCC, we compute the mean squared error to quantify absolute differences:

$$\text{MSE}_{k,m} = \frac{1}{M} \sum_{i=1}^{M} (P_k[i] - P_{(k+500m) \bmod N}[i])^2, \tag{17}$$

where $M$ denotes the total number of pixels in the BEV feature map.

**Perturbation-based Metrics.**   Let $x_c$ and $x_r$ denote the clean camera and radar inputs, and $x_c^{\text{pert}}$, $x_r^{\text{pert}}$ their respective perturbed versions. Let $F_{\text{camera}}(x_c, x_r)$ and $F_{\text{radar}}(x_c, x_r)$ denote the modality-specific predictions.

We define the following four metrics:

- $R_p/R$. Radar prediction change under radar perturbation:

    $$\text{PCC}_{k,m}(F_{\text{radar}}(x_c, x_r^{\text{pert}}), F_{\text{radar}}(x_c, x_r)), \quad \text{MSE}_{k,m}(F_{\text{radar}}(x_c, x_r^{\text{pert}}), F_{\text{radar}}(x_c, x_r))$$

    *Lower PCC and higher MSE are better*, indicating sensitivity to radar input.

- $R_p/C$. Camera prediction change under radar perturbation:

    $$\text{PCC}_{k,m}(F_{\text{camera}}(x_c, x_r^{\text{pert}}), F_{\text{camera}}(x_c, x_r)), \quad \text{MSE}_{k,m}(F_{\text{camera}}(x_c, x_r^{\text{pert}}), F_{\text{camera}}(x_c, x_r))$$

    *Higher PCC and lower MSE are better*, indicating invariance of the camera modality.

- $C_p/R$. Radar prediction change under camera perturbation:

    $$\text{PCC}_{k,m}(F_{\text{radar}}(x_c^{\text{pert}}, x_r), F_{\text{radar}}(x_c, x_r)), \quad \text{MSE}_{k,m}(F_{\text{radar}}(x_c^{\text{pert}}, x_r), F_{\text{radar}}(x_c, x_r))$$

    *Higher PCC and lower MSE are better*, indicating invariance of the radar modality.

- $C_p/C$. Camera prediction change under camera perturbation:

    $$\text{PCC}_{k,m}(F_{\text{camera}}(x_c^{\text{pert}}, x_r), F_{\text{camera}}(x_c, x_r)), \quad \text{MSE}_{k,m}(F_{\text{camera}}(x_c^{\text{pert}}, x_r), F_{\text{camera}}(x_c, x_r))$$

    *Lower PCC and higher MSE are better*, indicating sensitivity to camera input.

These metrics are averaged over the entire test set using uniformly sampled perturbations. Together, they verify whether the decomposition satisfies: (1) sensitivity to the perturbed modality and (2) invariance to the unperturbed modality — key indicators of successful modality separation.

# E   Further Discussion

## E.1   Broader Impact

LMD introduces the first post-hoc framework capable of attributing the predictions of each layer in a multi-sensor fusion model to its respective modalities, without modifying the original architecture.

**Positive societal impacts** may include:

- **Enhanced safety in autonomous systems.** By revealing the sensor modality (camera, LiDAR, or radar) primarily responsible for each prediction, LMD enables identification of single-sensor failures before they propagate into critical errors.

- **Accelerated certification and debugging.** Modality-level attributions can support system audits and facilitate the generation of safety evidence, in line with emerging regulatory requirements for ADAS systems.

- **Extension of interpretability to other multi-modal domains.** Since LMD is agnostic to model architecture, it can be applied to a wide range of multimodal perception systems beyond autonomous driving, such as medical image–text fusion or surveillance video analysis, to clearly attribute the role of each input modality in the model's decision-making process.

- **Commitment to open science.** We will publicly release our codebase, pretrained checkpoints, and evaluation scripts under an open-source license to promote reproducibility and rigorous external validation.

**Potential risks include:**

- **Misleading reassurance.** Users may over-trust visual explanations; a clear modality-wise attribution does not guarantee that the fused decision is accurate.

- **Bias amplification.** LMD uncovers but does not correct dataset imbalances, and naive use of its explanations may perpetuate structural biases.

- **Residual opacity.** The interpretability of high-order interactions and root-point selections in activation layers remains an open challenge for future work.

To mitigate these risks, we recommend pairing LMD with the perturbation-based metrics introduced in this paper to quantitatively assess explanation fidelity prior to deployment. We also encourage conducting independent reproducibility audits on geographically and demographically diverse datasets, and suggest applying the same level of scrutiny—augmented by domain-specific expertise and operational context—when extending LMD to other multimodal domains, such as medical imaging or service robotics. LMD sets a new benchmark for interpretability in multimodal domains and provides a solid foundation for future research and real-world deployment.

## E.2  Model-Wise Explanation

Understanding the extent to which we can perform model-wise explanations by comparing the camera-only model and the fusion model is a critical issue. Interpreting the contributions of each modality based on the results of these two models for specific data is highly challenging. This difficulty arises, for the two models are trained on different input distributions.

For instance, if a fusion model successfully detects an object that camera-only model fails to detect, it would be a clear error to attribute this success to radar data. Predictions from the camera-only model should be utilized only for references. For example, if the camera-only model successfully detects a particular vehicle, it can be inferred that the information captured by the camera sensor for that region is sufficient. This insight can be considered when analyzing the fusion model using the LMD approach.

Nonetheless, it is crucial to recognize that LMD is designed to provide post-hoc interpretations for a single trained model, not to be a tool for interpreting two different models trained on different data distributions.

## E.3  GradCAM Experiments

Since LMD enables access to internal activation maps, it is possible to compute activation maps from radar data and subsequently apply the Grad-CAM [36] approach.

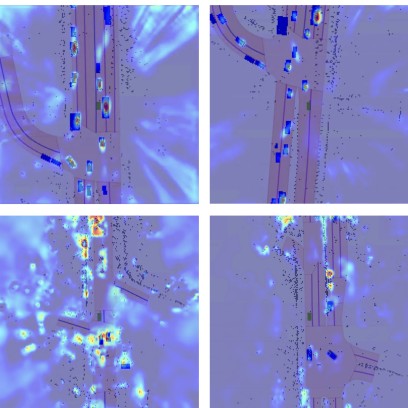

Figure 5: Visualizations of Radar-GradCAM Using Radar's Activations from LMD.

Grad-CAM [36] leverages gradient information to highlight relevant areas, Grad-CAM++ [37] introduces modified approach by taking positive gradients of target functions, and Score-CAM [38] bypasses the need for gradient information altogether, instead relying on activation scores to determine feature significance. Seg-Grad-CAM [39] is an adaptation of the Grad-CAM method, applied to the task of image segmentation by treating the segmentation output as the target function while maintaining the original mechanism of Grad-CAM for visual explanations. This approach enables a focused analysis of regions relevant to segmentation tasks, leveraging the gradients of the target output with respect to the feature maps. The Grad-CAM is calculated by :

$$L_{\text{Grad-CAM}}^c = \text{ReLU}\left(\sum_k \alpha_k^c A^{lk}\right) \tag{18}$$

where $L_{\text{Grad-CAM}}^c$ is the class activation map for class $c$, $\alpha_k^c$ are the weights for the $k$-th feature map in $l$-th layer $A^{lk}$, computed as the global average of the gradients flowing back from the output unit for class $c$. ReLU is

Table 11: Comparison of LRP and LMD methods.

| Method | Target | Result | Direction | Computation | Element-wise Op. | Property |
|--------|--------|--------|-----------|-------------|------------------|----------|
| LRP | Output Segment | Input Segment | Backward | 2 FWD + 1 BWD | Direct pass | Conservation rule |
| LMD | Input Segment | Output Segment | Forward | 2 FWD | Scaled pass | Eq.(8) |

applied to focus on features that have a positive influence on the class of interest. The weight of $k$-th feature map for class c is computed as follows.

$$\alpha_k^c = \frac{1}{Z} \sum_i \sum_j \frac{\partial y^c}{\partial A_{ij}^{lk}} \tag{19}$$

where $y^c$, $A_{ij}^{lk}$, and $Z$ denote the score for class c, $(i,j)$-th entry in $k$-th activation map, and normalization factor respectively. In Seg-Grad-CAM, $y^c$ is replaced by $\sum_{(i,j) \in R} y_{ij}^c$, where $R$ is a set of pixel indices of interest in the output mask.

Through obtained activations which are derived solely from the radar data with LMD, it becomes possible to use the conventional CAM method based on the activations at the intermediate layer, which we call Radar-GradCAM. The Radar-GradCAM is calculated by :

$$L_{\text{Radar-GradCAM}}^c = \text{ReLU}\left(\sum_k \alpha_k^c (\hat{A}^{lk}(\mathbf{o_c}, \mathbf{x_r}) - \hat{A}^{lk}(\mathbf{o_c}, \mathbf{o_r}))\right) \tag{20}$$

, where $\hat{A}^{lk}(\mathbf{o_c}, \mathbf{x_r})$ and $\hat{A}^{lk}(\mathbf{o_c}, \mathbf{o_r})$ are the activations of linearized fusion model $\hat{F}$ and $\alpha_k^c$ is calculated in the same manner as in eq. (19).

$$\alpha_k^c = \frac{1}{Z} \sum_i \sum_j \frac{\partial \sum_{(i,j) \in R} \hat{F}_j^l(\mathbf{x_c}, \mathbf{x_r})}{\partial A_{ij}^{lk}(\mathbf{x_c}, \mathbf{x_r})} \tag{21}$$

Based on this calculation, the Radar-GradCAM shown in Figure 5 reveals which parts of the fusion model are relevant to the radar data points, by taking the radar-only activations that are crucial to the perception task.

### E.4   Positioning LMD

In Table 11, we compare LMD with LRP to clarify its positioning. In LRP, for element-wise operations, such as normalization or activation layers, relevance scores are directly assigned to input neurons with distinct bias splitting rules [14, 16, 17, 11]. In contrast, LMD adheres to original function behavior by retaining the scale information used in element-wise operations. Also, LRP requires two forward passes and one backward pass with gradient checkpointing [40], while LMD requires one forward pass to store the information, allowing it to be implemented with a total of two forward passes. Lastly, LRP must satisfy the conservation rule during the backward pass to ensure that the output relevance score is correctly interpreted. In contrast, LMD follows the original function behavior, focusing on modality decomposition and adheres to Eq.(8). These different constraints arise from the distinct goals of each method.

## F   Extension to General Multimodal Model

In this section, we demonstrate that the LMD method can be generalized to multimodal models. We achieve this by extending the formulation to accommodate $M$ modalities.

Beginning at the fusion layer ($l = 1$), the first-order Taylor expansion of $f_j^1$ becomes :

$$f_j^1(\mathbf{x_{m_1}}, \dots, \mathbf{x_{m_M}}) = \sum_i \sum_{m \in \{m_1, \dots, m_M\}} J_{mji} x_{mi} + \\ \underbrace{\left(f_j^1(\tilde{x}_{m_1}, \dots, \tilde{x}_{m_M}) - \sum_i \sum_{m \in \{m_1, \dots, m_M\}} J_{mji} \tilde{x}_{mi} + \epsilon\right)}_{b_j^1} \tag{22}$$

where $m$ indexes the modality, $\tilde{x}_{m_1}, \dots, \tilde{x}_{m_M}$ represent reference points for $M$ modalities, and $\mathbf{J}_{mji}$ denotes the Jacobian calculated at the reference points for modality $m$.

For subsequent layer with $l = 2, \ldots, N$, suppose $l$-th layer function takes $\sum_{m \in \{m_1, \ldots, m_M, b\}} h_{mj}^{l-1}$ as inputs, the following holds :

$$f_j^l \left( \sum_{m \in \{m_1, \ldots, m_M, b\}} h_{mj}^{l-1} \right) =$$

$$\sum_i \sum_{m \in \{m_1, \ldots, m_M, b\}} J_{mji}^{l-1}(h_{mi}^{l-1}) + b_j^l = \sum_{m \in \{m_1, \ldots, m_M, b\}} h_{mj}^l \tag{23}$$

We define all modality features that all layers in fusion model be decomposed into them.

$$F_j^l(\mathbf{x_{m_1}}, \ldots, \mathbf{x_{m_M}}) = f_j^l \left( \sum_{m \in \{m_1, \ldots, m_M, b\}} h_{mj}^{l-1} \right) =$$

$$h_{m_1 j}^l + \cdots + h_{m_M j}^l + h_{bj}^l \tag{24}$$

From eq. (24), we can confirm inductively that the following holds. For the linearized layer functions $\hat{f}^1, \ldots, \hat{f}^N$ with bias-splitting rules applied, $F_j^l(\mathbf{x_{m_1}}, \ldots, \mathbf{x_{m_M}}) = \hat{F}_j^l(\mathbf{x_{m_1}}, \ldots, \mathbf{x_{m_M}})$ for $l \in \{1, \ldots, N\}$, where $\hat{F}^l(\mathbf{x_{m_1}}, \ldots, \mathbf{x_{m_M}})$ be $\hat{f}^l \circ \cdots \circ \hat{f}^2 \circ \hat{f}^1(\mathbf{x_{m_1}}, \ldots, \mathbf{x_{m_M}})$, forcing the epsilon in eq. (22) to zero. Based on all the layer functions, $\hat{f}^1, \ldots, \hat{f}^N$, the modality features are computed as follows :

$$h_{mj}^l = \hat{f}_j^l(h_{mj}^{l-1}), \ m \in \{m_1, \ldots, m_M, b\} \tag{25}$$

Furthermore, the following arithmetic is satisfied for layer function $\hat{f}_j^l$ :

$$\hat{f}_j^l \left( \sum_{m \in \{m_1, \ldots, m_M, b\}} h_{mj}^{l-1} \right)$$

$$= \hat{f}_j^l(h_{m_1 j}^{l-1}) + \cdots + \hat{f}_j^l(h_{m_M j}^{l-1}) + \hat{f}_j^l(h_{bj}^{l-1}) \tag{26}$$

From eq. (25) and eq. (26), the following property holds for every layers :

$$\hat{F}_j^l(\mathbf{x_{m_1}}, \ldots, \mathbf{x_{m_M}}) = \underbrace{\hat{F}_j^l(\mathbf{x_{m_1}}, \mathbf{o_{m_2}}, \ldots, \mathbf{o_{m_M}})}_{h_{m_1 j}^l}$$

$$+ \cdots + \underbrace{\hat{F}_j^l(\mathbf{o_{m_1}}, \ldots, \mathbf{o_{m_{M-1}}}, \mathbf{x_{m_M}})}_{h_{m_M j}^l} \tag{27}$$

$$+ \underbrace{\hat{F}_{m_M}^l(\mathbf{o_{m_1}}, \ldots, \mathbf{o_{m_M}})}_{h_{bj}^l}.$$

The linearization and bias-splitting rules applied to the normalization and activation layers are consistent with those described in the main paper.

