# OpenReview forum: "Layer-Wise Modality Decomposition for Interpretable Multimodal Sensor Fusion"
_NeurIPS.cc/2025/Conference — NeurIPS 2025 poster_

### Official Review · Reviewer_3nmi · 2025-07-01

**Clarity:** 3
**Significance:** 2
**Originality:** 3
**Rating:** 5
**Confidence:** 3

**Summary:**

The paper introduces Layer-Wise Modality Decomposition (LMD), a post-hoc interpretability method designed to disentangle modality-specific contributions in pretrained multimodal sensor fusion models for autonomous driving. LMD enables attribution of predictions to individual input modalities like camera and radar or camera and LIDAR across all layers without altering the model architecture.

The method locally linearises neural network operations to separate modality-specific features and bias, drawing inspiration from techniques like Layer-Wise Relevance Propagation (LRP) and Deep Taylor Decomposition (DTD). LMD’s effectiveness is evaluated through perturbation-based evaluations and visual decompositions by modality, showcasing its ability to clearly distinguish each modality’s role in a complex multimodal model.

Experiments show LMD successfully identifies modality-specific influences, with radar-based predictions responding strongly to radar input changes while camera-based predictions remain stable, and vice versa. While focused on autonomous driving, the paper highlights LMD’s applicability beyond that application, offering a scalable solution for interpreting complex multimodal perception models without imposing constraints on the architecture.

**Questions:**

- The paper does not discuss limitations or failure cases of LMD. This seems like a crucial missing element. Can you share scenarios where LMD might struggle (e.g., highly correlated modalities, extreme edge cases) and how you plan to address these in future work?
- Have you considered benchmarking LMD against other post-hoc interpretability methods (e.g. Grad-CAM, SHAP) to further validate its advantages in multimodal settings? This would really help to contextualise the work.
- The paper claims LMD’s applicability extends beyond autonomous driving (Page 1), and sensor fusion has much broader applicability beyond AV.  However, no specific examples or experiments in other domains are provided. Can you share potential applications in other fields (e.g., medical imaging, robotics) and any preliminary results or plans for such extensions? While it's not critical to address this, this could significantly expand the scope of the paper and contributions.

To summarise, in general the paper is lacking diversity in experiments performed, particularly in terms of exploring different models beyond SimpleBEV. If this were expanded to include other AV-related models and/or even better models in different domains, this could broaden the appeal and impact of this work, and I would raise my score.

**Ethical Concerns:**

["NO or VERY MINOR ethics concerns only"]

**Final Justification:**

Thanks to the authors detailed rebuttals and planned reviewers in response to all the reviewer concerns, I believe they have adequately addressed the concerns raised (including those I raised), and my final score reflects this. In particular, the additional experiments performed by the authors (e.g. LMD to CRN, and more than 2 modalities) were the most important in resolving my outstanding concerns leading to my raised score. It is therefore important that these revisions make it into the final, camera-ready version.

**Limitations:**

Yes

**Quality:**

2

**Strengths And Weaknesses:**

Clarity: The paper is well-structured, with clear sections that motivate the problem and introduce the proposed Layer-Wise Modality Decomposition (LMD) approach well. Technical concepts are explained with sufficient detail, supported by references to foundational works, and the contributions introduced are contextualised well relative to previous research. While code wasn't available for review, the promise of making code available for the model and experiments would further support the reproducibility, and enable researchers to more easily validate and build upon the method.

Experiments and evaluation: the paper provides validation of their method through the perturbation-based metrics and modality-wise visual decompositions (as seen in the evaluation of camera-radar and camera-LIDAR setting). The perturbation-based metrics and visualizations are well designed to test the method’s core claim of modality separation, and support LMD’s claims. However, comparisons with more baselines and broader modality testing could enhance credibility. In general, the core set of experiments chosen were sensible but quite limited in scope. Broadening this could significantly improve the strength of the paper.

Significance and Impact: The paper addresses a critical gap in interpretable AI for multimodal perception models, particularly in autonomous driving, where transparency in decision-making is vital for safety and reliability. The model-agnostic nature and applicability to high-capacity architectures make it broadly relevant, with potential impact beyond autonomous driving. On the other hand, the lack of non-driving examples and real-world integration details limits its demonstrated impact. In addition, there are limited modalities explored: the evaluation focuses on camera-radar and camera-LIDAR settings, but the paper does not discuss whether other modality combinations were tested, limiting the generality of conclusions that can be drawn.

---

> ### Author Rebuttal · Authors · 2025-07-31
>
> We sincerely thank the reviewer for the thoughtful and constructive feedback. We address your concerns as thoroughly as possible in the following reply.
>
> > **&nbsp; Comparisons with more baselines and broader modality testing could enhance credibility**
>
> Thank you for suggesting ways to strengthen the credibility of the paper. The results of extending LMD to three modalities can be found in our reply to ```Reviewer UztF```, the experiments combining SHAP with LMD are presented in our response to ```Reviewer KNAK```, and a conceptual comparison between LMD and other interpretability models is provided in our response to ```Reviewer opMU```.
>
> As you correctly pointed out the limited scope of evaluation, we will include the experiments covered in the rebuttal into the main paper for the camera-ready version.
>
>
> ## **Q1**
> > **&nbsp; The paper does not discuss limitations or failure cases of LMD. This seems like a crucial missing element. Can you share scenarios where LMD might struggle (e.g., highly correlated modalities, extreme edge cases) and how you plan to address these in future work?**
>
> We sincerely appreciate that the reviewer’s comments have allowed us to clarify the limitations of LMD and to outline directions for future work.
>
> (1) Failure cases of LMD.
> One limitation we observed is that while LMD provides an exact modality-wise decomposition for a pretrained model, practical interpretability requires a level of human-eye accessibility. Since LMD decomposes the pretrained model predictions into modality-specific outputs, the resulting maps contain noises which is difficult to interpret. We attribute this to the fact that the fusion model was never trained using single-modality prediction as an objective; hence, the model is not required to produce clean predictions for each modality individually.
>
> (2) Lack of benchmarks.
> To the best of our knowledge, no existing methods provide comparable interpretability for multimodal fusion outputs in autonomous driving scenarios. This makes it challenging to evaluate LMD against benchmarks. For example, the faithfulness scores [3] which is commonly adopted in domain are not directly applicable to our tasks. As part of our future work, we plan to explore and design rigorous ways to evaluate the faithfulness of LMD in multimodal perception tasks.
>
>
>
> ## **Q2**
> > **&nbsp; Have you considered benchmarking LMD against other post-hoc interpretability methods (e.g. Grad-CAM, SHAP) to further validate its advantages in multimodal settings? This would really help to contextualize the work.**
>
> In response to similar requests from multiple reviewers, we performed additional experiments incorporating SHAP [1] and attention-based attribution methods on transformer-based models. For detailed comparisons with SHAP and LRP, please refer to our responses to ```Reviewer KNAK``` and ```Reviewer opMu```.
>
>
> ## **Q3**
> > **&nbsp;  The paper is lacking diversity in experiments performed, particularly in terms of exploring different models beyond SimpleBEV. If this were expanded to include other AV-related models and/or even better models in different domains, this could broaden the appeal and impact of this work.**
>
> As many reviewers have pointed out the issue, we conducted additional experiments applying LMD to CRN [2] which is a camera-radar fusion model exploiting attention as a fusion operation. Specifically, the softmax operation in the attention mechanism was treated in the same manner as the activation layers described in the main paper.
>
> While we successfully decomposed modality-specific terms in CRN model and we observed that, aside from the constant bias prediction, high-order term arose due to matrix multiplication between input variables. In the camera-ready version, we will include the corresponding qualitative visualizations and a demonstration validating human-eye accessibility.
>
>
> ----------------------------------------------------------------------------------
> **Table A.** Comparison (Pearson Correlation) of LMD Variants with CRN [2]
>
> | **Modality**        | **Method**          | **Rₚ/R (↓)**        | **Rₚ/C (↑)**        | **Cₚ/R (↑)**        | **Cₚ/C (↓)**        |
> |---------------------|--------------------|---------------------|---------------------|---------------------|---------------------|
> | **Camera + Radar**  | Uniform - Identity | 0.6123 ± 0.043         | 1.0000 ± 0.000         | 1.0000 ± 0.000         | 0.5912 ± 0.052         |
> |                     | Identity - Identity| 0.2184 ± 0.125         | 1.0000 ± 0.000         | 1.0000 ± 0.000         | 0.3891 ± 0.043         |
> |                     | Identity - Uniform | 0.2572 ± 0.067         | 1.0000 ± 0.000         | 0.9882 ± 0.010         | 0.3827 ± 0.046         |
> |                     | Identity - Ratio   | **0.0552 ± 0.0213**         | **1.0000 ± 0.000**         | **1.0000 ± 0.000**         | **0.1571 ± 0.045**         |
>
> **Table B.** Comparison (Mean Squared Error) of LMD Variants with CRN [2]
> | **Modality**        | **Method**          | **Rₚ/R (↓)**        | **Rₚ/C (↑)**        | **Cₚ/R (↓)**        | **Cₚ/C (↑)**        |
> |---------------------|--------------------|---------------------|---------------------|---------------------|---------------------|
> | **Camera + Radar**  | Uniform - Identity | 11.5812 ± 2.712         | 0.0000 ± 0.000         | 0.0000 ± 0.000         | 39.4135 ± 19.802        |
> |                     | Identity - Identity| 10.5438 ± 1.714         | 0.0000 ± 0.000         | 0.0000 ± 0.000         | 28.8149 ± 18.801        |
> |                     | Identity - Uniform | 9.4676 ± 2.295         | 0.0000 ± 0.000         | 0.0087 ± 0.041         | 27.5582 ± 16.711        |
> |                     | Identity - Ratio   | **8.9732 ± 2.142**         | **0.0000 ± 0.000**         | **0.0000 ± 0.000**         | **54.4890 ± 22.151**        |
>
>
>
> ----------------------------------------------------------------------------------
> [1] Lundberg, Scott M., and Su-In Lee. "A unified approach to interpreting model predictions." Advances in neural information processing systems 30 (2017).
>
> [2] Kim, Youngseok, et al. "Crn: Camera radar net for accurate, robust, efficient 3d perception." Proceedings of the IEEE/CVF International Conference on Computer Vision. 2023.
>
> [3] Petsiuk, Vitali, Abir Das, and Kate Saenko. "Rise: Randomized input sampling for explanation of black-box models." arXiv preprint arXiv:1806.07421 (2018).

---

> > ### Comment · Reviewer_3nmi · 2025-08-05
> >
> > I thank the authors for their detailed rebuttals and additional results/planned reviewers in response to all the reviewer concerns. I believe they have adequately addressed most concerns, including those I raised, and I will increase my score to Accept in light of this.

---

### Official Review · Reviewer_wjuQ · 2025-07-02

**Clarity:** 1
**Significance:** 2
**Originality:** 2
**Rating:** 4
**Confidence:** 4

**Summary:**

This paper introduces Layer-Wise Modality Decomposition (LMD), a post-hoc interpretability method designed to attribute predictions of multimodal sensor fusion models in autonomous driving to individual input modalities. The paper introduces a method called LMD, designed for interpreting multimodal sensor fusion models in autonomous driving without altering the model architecture. The method is applied to camera-radar and camera-LiDAR fusion scenarios and assessed using structured perturbation-based metrics and visual decomposition approaches.

**Questions:**

1. The theoretical proof of LMD’s linearization is missing. This paper relies on first-order Taylor expansions to justify LMD’s decomposition, but it never formally proves that this approximation maintains the claimed properties across all layers. If LMD is to be trusted in safety-critical systems, the authors must provide formal guarantees—not just empirical observations. Otherwise, the entire theoretical foundation is not solid.
2. The proposed LMD should be compared against existing well-known solutions like SHAP, LRP, fANOVA or Grad-CAM.
3. The term "model-agnostic" should be clarified. More details should be provided to demonstrate its importance. In this paper, the proposed solution is only evaluated on camera-radar and camera-LiDAR models, but more experiments should be provided to demonstrate its generality.
4. The legends of green markers in Figure 2 are missing, and more quantitative validation should be provided.
5. There is no analysis of LMD’s behavior under adversarial or noisy inputs. Autonomous driving sensors frequently encounter noise, occlusions, and adversarial conditions. If radar data is corrupted, does LMD’s decomposition degrade gracefully, or does it produce nonsensical attributions?
6. If camera and radar disagree (e.g., one detects an obstacle, the other doesn’t), how does LMD attribute the final prediction? Does it highlight the conflict, or does it silently favor one modality?
7. The computational overhead of LMD should be discussed. The low latency is important for the safety-critical autonomous system.
8. The authors point out that root point selection and high-order interactions in activation layers are unresolved, yet how to solve them in LMD is not pending.
9. How to select the reference point for the Taylor expansion should be explained.
10. Are the results statistically significant?

**Ethical Concerns:**

["NO or VERY MINOR ethics concerns only"]

**Final Justification:**

The authors have provided more expriments about the model on three-modality setting.
The key contribution of this paper is highlighted in the revison.
I'm willing to update the raking of this paper. However, I am still concerning of the generality of the proposed model for more than three modalities.

**Limitations:**

The authors do not adequately address the limitations of their work. For example, they do not discuss the computational cost of LMD or how it scales with increasing model size or number of modalities.They do not analyze the sensitivity of LMD to hyperparameter choices or architectural variations. They fail to consider the potential misuse of the method in safety-critical systems where incorrect interpretations could lead to dangerous decisions.

**Quality:**

2

**Strengths And Weaknesses:**

Strengths

1: The novel model-agnostic approach LMD is proposed to decompose contributions of individual modalities in pretrained fusion models. The research problem in this paper is important for numerous downstream applications.

2. The linearization of activation and normalization layers is mathematically formulated.

3. The proposed solution is broadly evaluated across across camera-radar and camera-LiDAR configurations to demonstrate the generality.
 solution is


Weaknesses

1. The paper relies on first-order Taylor expansions to linearize non-linear layers (e.g., activation and normalization layers), but it does not rigorously justify why this approximation is sufficient or how high-order interactions are handled. More details of the theoretical foundation for LMD should be provided.

2. The decomposition process may introduce significant computational costs. The runtime and memory overhead analysis should be provided.

3. The proposed solution is not evaluated under realistic conditions. However, it is important for performance evaluation under noisy or adversarial inputs to demonstrate the efficiency.

4. The presentatoin of this paper should be improved to highlight the key contributions.

---

> ### Author Rebuttal · Authors · 2025-07-31
>
> We sincerely thank you for thorough evaluation and constructive feedback. Your comments provide valuable insights that help us clarify the theoretical foundation of LMD.
>
> ## **W1, Q8**
> > **&nbsp; The paper relies on first-order Taylor expansions to linearize non-linear layers (e.g., activation and normalization layers), but it does not rigorously justify why this approximation is sufficient or how high-order interactions are handled. More details of the theoretical foundation for LMD should be provided.**
>
> Thank you for raising a key question closely related to the core of our work.
> Please check our responses to ```Reviewer UztF```.
>
> ## **Q1**
> > **&nbsp; The theoretical proof of LMD’s linearization is missing.**
>
> We appreciate the reviewer’s emphasis on formal guarantees.
>
> We believe the paper includes the theoretical proofs. If you could specify which parts are perceived as missing, we'll clarify the ambiguous points and include the corresponding details in the main paper.
>
>
> ## **Q2**
> > **&nbsp; The proposed LMD should be compared against existing well-known solutions like SHAP, LRP, fANOVA or Grad-CAM.**
>
> We believe this point is crucial for demonstrating the validity and efficiency of our proposed method.
> Please refer to our responses to ```Reviewer KNAK``` and ```Reviewer opMu```.
>
>
> ## **W2, Q7**
> > **&nbsp; The decomposition process may introduce significant computational costs. The runtime and memory overhead analysis should be provided.**
>
> We appreciate the reviewer for highlighting these important concerns.
> Please check our responses to ```Reviewer opMu```.
>
>
> ## **Q3**
> > **&nbsp; More details should be provided to demonstrate its importance. In this paper, the proposed solution is only evaluated on camera-radar and camera-LiDAR models, but more experiments should be provided to demonstrate its generality.**
>
> We have addressed your concern along with those raised by other reviewers by conducting additional experiments of LMD on 3-modalities fusion and attention-based models. For detailed results and analysis, please refer to our response to ```Reviewer UztF``` and ```Reviewer 3nmi```.
>
>
> ## **Q4**
> > **&nbsp; The legends of green markers in Figure 2 are missing.**
>
> To the best of our knowledge, the green markers are correctly included in the figure. If you could clarify which part might have caused the misunderstanding, we would be happy to address it and make the necessary adjustments.
>
>
> ## **Q5**
> > **&nbsp; There is no analysis of LMD’s behavior under adversarial or noisy inputs. Autonomous driving sensors frequently encounter noise, occlusions, and adversarial conditions. If radar data is corrupted, does LMD’s decomposition degrade gracefully, or does it produce nonsensical attributions?**
>
> We understand your concern, as this aspect is indeed critical for typical explanation methods. However, our proposed LMD provides an exact description of the pretrained model’s behavior. Therefore, any vulnerability to adversarial or noisy inputs reflects the vulnerability of the pretrained model rather than a weakness of the LMD itself. If you have a different perspective on this point, we would appreciate further clarification and are open to addressing it.
>
>
> ## **Q6**
> > **&nbsp; If camera and radar disagree (e.g., one detects an obstacle, the other doesn’t), how does LMD attribute the final prediction? Does it highlight the conflict, or does it silently favor one modality?**
>
> Figure.2 in our paper specifically illustrates how LMD interprets the model predictions under such conditions. As with Q5, it is important to note that LMD is not an explanation model but an exact description of the pretrained model, so it does not introduce  biases of favoring a particular modality, which can be a concern in surrogate-model (such as LIME and Kernel-SHAP). If this was not the point you intended to raise, we would greatly appreciate clarification.
>
>
> ## **Q9**
> > **&nbsp; How to select the reference point for the Taylor expansion should be explained.**
>
> To the best of our understanding, the paper does address how the Jacobian is used in relation to the reference point. For activation layers, instead of computing the Jacobian at a specific reference point, we set the diagonal entries of Jacobian to the slope of the line or segment connecting the two operating points to satisfy LMD's constraints. For LayerNorm, computing the Jacobian is not meaningful; therefore, we use a scaling factor as a constant, with the justification provided in the paper. If any aspects of the paper were unclear, we would be happy to take them into consideration accordingly.
>
>
> ## **Q10**
> > **&nbsp; Are the results statistically significant?**
>
> Thank you for posing a fundamental question regarding our work. The reason why the results statistically significant is that the fact that different modalities are not affected by modality perturbation indicates that we can have strong confidence in the interpretation of modality attribution for the pretrained model's predictions. In other words, the statistically confirmed separation ensures that each modality contribution can be trusted as an isolated interpretation, providing a robust foundation for modality attribution in safety-critical perception systems.

---

> > ### Comment · Reviewer_wjuQ · 2025-08-04
> >
> > Thanks for your response. Two problems are still pending.
> > For Q 5, it is important for autonmous driving. More experiment should be conducted to demonstrate the efficiecy of LMD when single or two modalities are imparied by the noise under adverse scenarios, and provide explainable visualization results to showcase why one specificed modality is used for the final decision.
> >
> > Additionaly, existing experiments mainly foucs on models with two modailities. How to extend LMD to models with more than two modalities is important.

---

> > > ### Author Response · Authors · 2025-08-05
> > >
> > > ## **Additional Q1.**
> > >
> > > Thank you for the clarification—this helped us better understand the intent behind Question 5.
> > >
> > > We now believe the core of the question concerns whether LMD can still provide meaningful interpretation in the presence of sensor failure, or whether LMD itself would also fail under such conditions.
> > >
> > > As shown in the main paper, quantitative results demonstrate that when one sensor modality becomes unreliable, the predictions based on other modalities remain unaffected. The separability stems from the LMD formulation and is supported by quantitative results presented in the paper.
> > >
> > > > **&nbsp; More experiment should be conducted to demonstrate the efficiecy of LMD when single or two modalities are imparied by the noise under adverse scenarios, and provide explainable visualization results to showcase why one specificed modality is used for the final decision.**
> > >
> > >
> > > Your question seems to go a step further—focusing on *qualitative analysis*, and more specifically on whether LMD can help *visually interpret or diagnose* such failures in human-eye accessible way.
> > >
> > > To explore this, we conducted an additional experiment using the dataset from [1], where the model was trained under sensor failure scenarios (e.g. 'camera lens occlusion', 'no LiDAR point inputs' or 'LiDAR object failure') to learn to rely on a single modality data in perception tasks. To assess the outcome, we visualized the modality-specific predictions produced by LMD.
> > >
> > >
> > > The qualitative results revelaed that corrupted sensor-based predictions became unreliable or meaningless, while those from the intact modalities remain interpretable. Specifically, in regions affected by camera occlusion, the model failed to produce camera-based predictions and instead relied on the LiDAR modality. These visualizations demonstrate the model's ability to fall back on unaffected modalities when one fails, thereby validating LMD's potential for interpretable multimodal failure analysis.
> > >
> > > In response to your insightful comment, we will include a discussion of this point in the main paper and provide the relevant visualizations.
> > >
> > >
> > > ----------
> > > ## **Additional Q2.**
> > > > **&nbsp; Additionaly, existing experiments mainly foucs on models with two modailities. How to extend LMD to models with more than two modalities is important.**
> > >
> > > We interpret the question as having two possible intents and will provide responses accordingly:
> > >
> > > **1. How can LMD be generally implemented for more than two modalities?**
> > >
> > > As stated in the main paper (line 143), general LMD formulation with $M$ modalities is provided in appendix F. The linearization methods are identical to those in two modalities setting in the main paper.
> > >
> > >
> > > **2. How did we extend the two-modality experiments to our setting with three modalities?**
> > >
> > > In our response to ```Reviewer UztF```, we conducted experiments using three modalities. Specifically, we used SimpleBEV model [2] as in the main paper, modifying it to fuse features from the three modalities (with the same fusion operation). We then trained the model, applied LMD accordingly.
> > >
> > >
> > > --------------
> > >
> > >
> > > We hope that all your questions have been sufficiently addressed. Please let us know if anything remains unclear. Again, thank you for your insightful comments.
> > >
> > > ---------------
> > > [1] Yu, Kaicheng, et al. "Benchmarking the robustness of lidar-camera fusion for 3d object detection." Proceedings of the IEEE/CVF Conference on Computer Vision and Pattern Recognition. 2023.
> > >
> > > [2] Harley, Adam W., et al. "Simple-bev: What really matters for multi-sensor bev perception?." ICRA 2023.

---

### Official Review · Reviewer_opMu · 2025-07-03

**Clarity:** 2
**Significance:** 4
**Originality:** 3
**Rating:** 4
**Confidence:** 2

**Summary:**

This paper introduces a method called Layer-Wise Modality Decomposition (LMD) to help understand how different sensors (like cameras, radar, and LiDAR) each contribute to a deep learning model’s prediction in autonomous driving. The method works after the model is trained and does not change the model itself. It breaks down the model’s output layer by layer, showing how much each sensor influenced the result. The authors test this method on real sensor fusion models and show that it can clearly separate the contributions from each sensor, helping make these systems more transparent and interpretable.

**Questions:**

I am leaning toward the negative side due to the major and minor concerns raised below.
1. Major question: Can you include or at least discuss comparing LMD with standard interpretability methods such as SHAP, or Grad-CAM in your sensor fusion setting? If these methods are not directly applicable, could you clarify why and possibly provide qualitative or conceptual comparisons?
2. Minor question: Can you provide a more detailed justification for your treatment of normalization layers (e.g., the use of Identity and Ratio rules)? These rules appear to be heuristic fixes to maintain LRP conservation principles—are there theoretical insights?
3. (Optional) Can you report on the computational efficiency of your method, such as runtime, memory usage, or scalability to larger models or datasets? This would help assess the practicality of LMD in real-world deployment scenarios.

**Ethical Concerns:**

["NO or VERY MINOR ethics concerns only"]

**Final Justification:**

I thank the authors for their additional explanations and new experiments addressing computational and memory complexity, as well as comparisons with SHAP.  The added experiments and clearer conceptual positioning had a positive impact on my evaluation, and I have accordingly raised my score.

**Limitations:**

The paper briefly acknowledges some limitations in the Discussion section, such as the lack of a thorough investigation into the choice of root points and high-order interactions between modalities. However, several important limitations remain insufficiently discussed:  The assumption of modality independence is strong and may not hold in real-world sensor fusion systems, but this is not explicitly addressed. The method’s feasibility for real-time or large-scale deployment is not discussed.

**Quality:**

2

**Strengths And Weaknesses:**

Strengths: (1) The paper presents a novel idea by extending Layer-Wise Relevance Propagation (LRP) and Deep Taylor Decomposition (DTD) to multimodal sensor fusion, enabling layer-wise decomposition by modality—a problem not well addressed in previous work. (2) The proposed method addresses an important gap in interpreting deep learning models used in safety-critical systems like autonomous driving. (3) LMD works post-hoc without modifying the original model, making it practical for real-world systems already in deployment. (4) The paper introduces structured perturbation-based metrics to measure modality separation, and tests the method across both radar-camera and LiDAR-camera setups.

Weaknesses: (1) Major Concern – Please correct me if I’m wrong—I tried my best to understand this point. The LMD method builds upon Deep Taylor Decomposition (DTD) as its core mechanism for relevance propagation. However, as shown in recent rigorous analysis (e.g., Sixt et al., 2022) [1], DTD often reduces to a simple gradient × input formulation under common settings, such as ReLU activations and typical network architectures. This suggests that in practice, DTD (and methods based on it like LMD) may not offer significantly richer or more faithful attributions than simpler gradient-based methods.  (2) Minor Concern –The linearization for normalization layers (BatchNorm, LayerNorm) breaks the relevance-conservation principle that standard LRP relies on. The paper proposes work-arounds, e.g., the identity rule, the ratio rule etc, which seem to restore conservation empirically. However, these fixes are presented as engineering heuristics; the paper does not offer a formal proof.  (3) Major Concern –The experiments lack standard interpretability benchmarks, such as comparisons with SHAP [2] or Grad-CAM [3].

[1] https://keonly.github.io/cards/Layer-wise-relevance-propagation
[2] Lundberg, S. M., & Lee, S.-I. (2017). A Unified Approach to Interpreting Model Predictions. NeurIPS.
[3] Selvaraju et al. (2017). Grad-CAM: Visual Explanations from Deep Networks via Gradient-Based Localization. ICCV.

---

> ### Author Rebuttal · Authors · 2025-07-31
>
> We sincerely thank you for the careful reading and the constructive feedback. Below we address every point in the order raised and describe the additions that will appear in the revised manuscript.
>
> ## **W1**
> > **&nbsp; LMD (based on DTD) may not offer significantly richer or more faithful attributions than simpler gradient-based methods.**
>
> As you correctly pointed out, both LRP and Deep Taylor Decomposition (DTD) can be reduced to an input × gradient formulation, particularly when the root point is chosen with the operating point, also discussed in [1].
>
> However, simple gradient-based methods confront significant limitations when applied to tasks involving high-dimensional inputs and outputs (e.g., perception tasks). In such settings, the computational complexity of computing Jacobians poses challenges applying the methods.
>
> Moreover, as discussed in the main paper, operations such as Layer Normalization present additional challenges: computing gradients through these layers fail to attribute function outputs to input features. This severely hinders interpretability and necessitates alternative strategies that preserve explanatory power while bypassing these limitations. The constraints (e.g., conservation property in LRP) and variants introduced in methods such as LRP and LMD are designed to effectively mitigate these challenges.
>
>
> ## **W2, Q2**
> > **&nbsp; The linearization for normalization layers (BatchNorm, LayerNorm) breaks the relevance-conservation principle that standard LRP relies on. The paper proposes work-arounds, e.g., the identity rule, the ratio rule etc, which seem to restore conservation empirically. However, these fixes are presented as engineering heuristics; the paper does not offer a formal proof.**
>
> Thank you for highlighting the critical aspect. The identity and uniform rules follow directly from Deep Taylor Decomposition (DTD) and from the well-known LRP treatments of renormalization layers. We adopted these rules as LMD variants for comparison experiments since they satisfy our constraints (eq. (7) in our paper). The ratio rule is not a heuristic; it's a closed-form solution to a constrained problem while preserving our assumptions.
>
>
>
> ## **W3 & Q1**
> > **&nbsp; The experiments lack standard interpretability benchmarks, such as comparisons with SHAP or Grad-CAM. Could you clarify conceptual comparisons?**
>
> In this reply, we provide a conceptual comparisons between LRP and our proposed LMD method. In LRP, for layers performing element-wise operations such as normalization or activation layers, relevance scores are directly mapped to input neurons, with only the bias splitting rule applied. However, our LMD method adheres closely to the original function behavior by retaining the scale information used in element-wise operations, resulting in disregard of the conservation rule. Additionally, since LRP is a backpropagation-based method, it requires gradient checkpointing, thus implemented with two forward passes and one backward pass. In contrast, LMD requires only one forward pass to store the information for, allowing it to be implemented with a total of two forward passes. Finally, LRP satisfies the conservation rule during the backward pass to ensure that the output relevance score is correctly interpreted. In contrast, LMD follows the behavior of the original function and is focused on modality decomposition. These differences in rules originate from the distinct goals of each method.
>
> -------------------------------------------------------------------------------------------------------------------------------------------------------------------------------
> **Table A.** Comparison of LRP and LMD method
> | **Method** | **Target**        | **Result**       | **Direction** | **Passes**       | **Element‑Wise Operation** | **Constraint**       |
> |:----------:|:----------------:|:----------------:|:-------------:|:---------------------:|:--------------------------:|:------------------:|
> | **LRP**    | Output Segment   | Input Segment    | Backward      | 2 FWD, 1 BWD | Direct Pass                | Conservation Rule  |
> | **LMD**    | Input Segment    | Out Segment      | Forward       | 2 FWD             | Scaled Pass                | Eq. (7)       |
>
> -------------------------------------------------------------------------------------------------------------------------------------------------------------------------------
>
>
>
> * FWD = forward pass, BWD = backward pass.
>
>
> -------------------------------------------------------------------------------------------------------------------------------------------------------------------------------
>
> Also, for a comparison with SHAP and our method, we suggest to check our response to ```Reviewer KNAK```.
>
>
> ## **Q3**
> > **&nbsp; Can you report on the computational efficiency of your method, such as runtime, memory usage, or scalability to larger models or datasets? This would help assess the practicality of LMD in real-world deployment scenarios.**
>
>
> We are pleased to provide further details regarding the computational and memory efficiency of our method.
>
>
> -------------------------------------------------------------------------------------------------------------------------------------------------------------------------------
> **Table B.** Computational and memory complexity of LMD, LRP-based and linear-time perturbation methods measured in a single forward pass.
>
> | Method                                   | Passes          | Computational Complexity | Memory Consumption |
> | ---------------------------------------- | ----------------- | ---------------------- | ----------- |
> | **LRP** (gradient checkpointing)         | 2 FWD + 1 BWD | O(1)                   | O(√ Nₗ)     |
> | **Shapley-based** | 2^M FWD            | O(2^M)                  | O(1)        |
> | **LMD**                                  | 2 FWD   | O(1)                   | O(1)        |
>
> -------------------------------------------------------------------------------------------------------------------------------------------------------------------------------
>
> * FWD = forward pass, BWD = backward pass.
> M = number of modalities,
> Nₗ = number of layers.
>
> You can check computational and memory efficiency of our methods from Table B. LRP incurs one backward pass and √ Nₗ memory via checkpointing. Shapley-based methods remain low memory consumption but scale exponentially with multiple modalities. LMD is efficient in terms of both computational complexity and memory consumption: it only requires two forward passes, making it computationally lightweight. Also LMD stores the necessary intermediate information at specific layers and immediately discards it after decomposition while keeping the overall memory usage low.
>
> Considering the issue was raised by multiple reviewers, We will incorporate it into the main paper to enhance the overall contribution.
>
> -------------------------------------------------------------------------------------------------------------------------------------------------------------------------------
> [1] Achtibat, Reduan, et al. "Attnlrp: attention-aware layer-wise relevance propagation for transformers." ICML 2024.

---

> > ### Comment · Reviewer_opMu · 2025-08-04
> >
> > Thanks for the additional explanations and experiments regarding computational and memory complexity, as well as the comparisons with SHAP. I also reviewed the new results alongside those in the original paper, and it appears that LMD consistently remains the most effective method. You might consider presenting these comparisons more closely together in the paper to highlight this point. Those clarifications have helped me better understand the work, and I have updated my score accordingly. Thank you again for your time and efforts.

---

> > > ### Author Response · Authors · 2025-08-04
> > >
> > > We appreciate your positive feedback on the additional experiments and clarifications. We will revise the manuscript to present the comparisons between LMD and recent interpretable methods more closely together, as you suggested.
> > > Thank you again for your valuable comments and for updating your score.

---

### Official Review · Reviewer_KNAK · 2025-07-03

**Clarity:** 3
**Significance:** 2
**Originality:** 3
**Rating:** 5
**Confidence:** 4

**Summary:**

The authors propose a post-hoc interpretability method for perception models in autonomous driving, which process multimodal inputs (image, LiDAR or radar data). Their method locally linearizes learned operations to separate modality-specific information and attribute the impact on model outputs.

**Questions:**

- As mentioned in the weaknesses, can you apply your method to another model and combine it with a feature-based method?
- Can you briefly discuss how your method relates to methods for mechanistic interpretability?
- Can you measure non-linear correlation between corrupted and non-corrupted forward passes, e.g. using the xicor metric [4]? This could allow for a more detailed comparison.
[4] Chatterjee, S., 2019. A new coefficient of correlation. arXiv e-prints, page. arXiv preprint arXiv:1909.10140, 711.

**Ethical Concerns:**

["NO or VERY MINOR ethics concerns only"]

**Final Justification:**

Thank you for the rebuttal.

The additional evidence using an additional model is well appreciated.
Results using SHAP as well Combination of LMD with SHAP are convincing.
More emphasis on a discussion of recent work In particular, recent mechanistic interpretability methods, which are also used to analyze motion forecasting models for autonomous driving would improve the manuscript.

I am ready to raise quality to 3 good now and go to accept.

**Limitations:**

yes

**Quality:**

2

**Strengths And Weaknesses:**

Strengths:
- The proposed method splits the processing of a multimodal model and generates predictions using each modality individually. This allows for comparing predictions based on fused features with predictions based on modality-specific features, which is a step toward interpreting the modality-specific contributions.
- The authors further show that their method disentangles fused features by corrupting one input modality and measuring no impact on the predictions based on the other modality. They perform this experiment for a multimodal model that processes either image and LiDAR or image and radar data. This should be elaborated before publication.
- The authors ablate different linearization approaches for batch, layer and instance norm layers. They measure the Pearson correlation and MSE between predictions from corrupted and non-corrupted forward passes.

Weaknesses:
- As a minor issue, while their method seems to be model-agnostic, the authors only evaluate it using one model. Experiments with further models would strengthen the paper. Nevertheless, this issue should at least be discussed in the final contribution.
- The proposed method is a step toward interpreting the impact of all features per processed modality, but doesn’t quantify or interpret the impact of specific features per modality. Combining the proposed method with a feature-based method like SHAP [1] would significantly strengthen the contribution.
- As a major shortcoming, the related work subsection on interpretation methods focuses on fairly old methods and doesn’t include recent mechanistic interpretability methods (see [3]), which are also used to analyze motion forecasting models for autonomous driving [4].

[1] Lundberg, S.M. and Lee, S.I., 2017. A unified approach to interpreting model predictions. Advances in neural information processing systems, 30.
[2] Bereska, L. and Gavves, S., Mechanistic Interpretability for AI Safety-A Review. Transactions on Machine Learning Research.
[3] Tas, O.S. and Wagner, R., 2024. Words in Motion: Extracting Interpretable Control Vectors for Motion Transformers. arXiv preprint arXiv:2406.11624.

The reported strengths make the paper generally interesting and publishable, assuming that the shortcomings will be resolved as outlined above.

---

> ### Author Rebuttal · Authors · 2025-07-31
>
> Thank you for the thoughtful constructive feedback and for highlighting the paper’s strengths.
> Below we address every concern raised and summarize the concrete additions we will include in the camera-ready.
>
>
> ## **W1 & Q1**
> > **&nbsp; While the method seems to be model-agnostic, the authors only evaluate it using one model. Experiments with further models would strengthen the paper. Nevertheless, this issue should at least be discussed in the final contribution.**
>
> Please check our responses to ```Reviewer 3nmi```, which contain results on models beyond SimpleBEV, incorporating transformer-based architectures to our method. As you suggested, we will address this issue by explicitly discussing it as part of our contributions.
>
>
> ## **W2**
> > **&nbsp; Combining the proposed method with a feature-based method like SHAP would significantly strengthen the contribution. Can you apply your method to another model and combine it with a feature-based method?**
>
>
> We sincerely thank you for proposing the interesting idea. In response to your suggestion (slightly different from what you suggested), we conducted experiments combining SHAP to our method. Specifically, we treated each modality as a distinct feature in the SHAP framework—for example, in a fusion model incorporating Camera, Radar, and LiDAR, we assumed three input features corresponding to each modality. We computed the full set of Shapley values over all possible coalitions. To estimate each sensor-based prediction, we followed the standard SHAP procedure: we randomly sampled data from the training set to replace the modality of interest, fed it into the model, and subtracted the resulting prediction from that of the original model.
>
> As noted in [1], SHAP is a local explanation method that assumes feature independence. To evaluate the stability of the SHAP-based explanation under this assumption, we performed the same modality replacement experiment as in our main paper and measured the correlation between the resulting predictions across different modalities. This allowed us to quantify how consistent the SHAP explanation remains in the presence of modality interactions.
>
> Additionally, we explored an approach that combines LMD with SHAP. Specifically, we first used LMD to decompose the model’s prediction into modality-based components. We then applied SHAP to the bias prediction to redistribute it back across the modality-specific predictions. This approach allows us to refine the attribution by leveraging the strengths of both methods.
>
> We conducted the modality-replacement experiment using this combined method observing that it better satisfied the assumption of modality-based separability compared to SHAP. Please refer to the table below for the quantitative results.
>
>
> -------------------------------------------------------------------------------------------------------------------------------------------------------------------------------
> **Table A.** Stability Experiments on SHAP and LMD + SHAP (Pearson Correlation)
> | **Modality** | **Metric (↑)** | **SHAP (Corr)** | **LMD + SHAP (Corr)** |
> |--------------|------------|----------------------|--------------------------|
> | **Camera-Radar (2-modal)** | Rₚ/C| 0.6909 ± 0.1119 | 0.9385 ± 0.0412 |
> |              | Cₚ/R | 0.6746 ± 0.0727 | 0.8942 ± 0.0472 |
> | **Camera-LiDAR (2-modal)** | Lₚ/C | 0.7062 ± 0.1172 | 0.9210 ± 0.0422 |
> |              | Cₚ/L      | 0.7205 ± 0.0591 | 0.9095 ± 0.0200 |
> | **Camera-Radar-LiDAR (3-modal)** | RₚLₚ/C | 0.7108 ± 0.1145 | 0.8790 ± 0.0522 |
> |              | CₚLₚ/R      | 0.7066 ± 0.0769 | 0.9574 ± 0.0099 |
> |              |  CₚRₚ/L      | 0.7025 ± 0.0685 | 0.9389 ± 0.0161 |
>
> -------------------------------------------------------------------------------------------------------------------------------------------------------------------------------
> **Table B.** Stability Experiments on SHAP and LMD + SHAP (Mean Squared Error)
> | **Modality** | **Metric (↓)** | **SHAP (MSE)** | **LMD + SHAP (MSE)** |
> |--------------|------------|--------------|------------------|
> | **Camera-Radar (2-modal)** | Rₚ/C | 0.0854 ± 0.0252 | 0.0059 ± 0.0042 |
> |              | Cₚ/R | 0.0191 ± 0.0029 | 0.0136 ± 0.0020 |
> | **Camera-LiDAR (2-modal)** | Lₚ/C | 0.0761 ± 0.0230 | 0.0094 ± 0.0062 |
> |              | Cₚ/L  | 0.0279 ± 0.0038 | 0.0181 ± 0.0028 |
> | **Camera-Radar-LiDAR (3-modal)** | RₚLₚ/C | 0.0651 ± 0.0186 | 0.0375 ± 0.0129 |
> |              | CₚLₚ/R | 0.0073 ± 0.0013 | 0.0031 ± 0.0007 |
> |              | CₚRₚ/L  | 0.0207 ± 0.0031 | 0.0101 ± 0.0017 |
>
> -------------------------------------------------------------------------------------------------------------------------------------------------------------------------------
>
>
> ## **W3, Q2**
> > **&nbsp; The related work subsection on interpretation methods focuses on fairly old methods and doesn’t include recent mechanistic interpretability methods. Can you briefly discuss how your method relates to methods for mechanistic interpretability?**
>
> Given that this was identified as a major shortcoming, we will provide a more detailed discussion on the connection between our approach and recent advances in mechanistic interpretability.
>
>
>
> -------------------------------------------------------------------------------------------------------------------------------------------------------------------------------
>
>
> ## **Q3**
> > **&nbsp; Can you measure non-linear correlation between corrupted and non-corrupted forward passes, e.g. using the xicor metric ? This could allow for a more detailed comparison.**
>
> We sincerely thank you for the valuable suggestion. However, due to time constraints during the rebuttal period, we were unable to conduct the proposed experiment.
>
>
>
> [1] Lundberg, Scott M., and Su-In Lee. "A unified approach to interpreting model predictions." Advances in neural information processing systems 30 (2017).

---

> > ### Comment · Reviewer_KNAK · 2025-08-01
> > **Thorough rebuttal with additional evidence.**
> >
> > Thank you for the rebuttal.
> > + The additional evidence using an additional model is well appreciated.
> > + Results using SHAP as well Combination of LMD with SHAP are convincing.
> > + More emphasis on a discussion of recent work In particular, recent mechanistic interpretability methods, which are also used to analyze motion forecasting models for autonomous driving would improve the manuscript.
> >
> > I am ready to raise quality to 3 good now.

---

> > > ### Author Response · Authors · 2025-08-03
> > >
> > > We appreciate your encouraging feedback and your decision to raise  the quality score.
> > >
> > > We fully agree elaborating a connection to mechanistic interpretability methods would enhance our paper. As you suggested, we will revise the related work section to incorporate a discussion of recent developments in mechanistic interpretability.
> > >
> > > Again, we sincerely appreciate your constructive guidance and updated assessment.

---

### Official Review · Reviewer_UztF · 2025-07-03

**Clarity:** 3
**Significance:** 4
**Originality:** 3
**Rating:** 4
**Confidence:** 2

**Summary:**

This paper introduces Layer-Wise Modality Decomposition (LMD), a post-hoc interpretability method designed to disentangle and attribute the contributions of individual sensor modalities (such as camera, radar, and LiDAR) across all layers of multimodal perception models, particularly in autonomous driving. Motivated by the opacity of high-capacity fusion models, LMD locally linearizes neural network layers to decompose their outputs into modality-specific components without altering the original architecture.

**Questions:**

1. Have the authors tested LMD on settings with more than two modalities? While Appendix F theoretically extends the method to an arbitrary number of modalities, it remains unclear whether any practical challenges arise in such cases, for example, in the selection of reference points or maintaining modality separation when scaling beyond two modalities.

2. Can the proposed method be readily applied to more complex fusion architectures, such as those incorporating cross-modal attention mechanisms — for example, the network used in TransFuser [1]?

3. Figure 2 shows "perceptual capability" from the bias term, which is described as representing constant offsets. However, what exactly does this component capture, residual cross-modal interactions, constant activations, or something else? If it truly represents constant information, does the bias-based prediction remain stable or show minor variations when presented with similar inputs?

[1] TransFuser: Imitation with Transformer-Based Sensor Fusion for Autonomous Driving, TPAMI 2023.

**Ethical Concerns:**

["NO or VERY MINOR ethics concerns only"]

**Final Justification:**

After reading the rebuttal and reviewing the authors' responses to other reviewers, I believe this work has demonstrated its value from a practical usage perspective. Therefore, I have decided to raise my score. I hope the authors will include the additional experiments in the revised version of the paper, as they provide essential evidence of the advantages of LMD.

**Limitations:**

yes

**Quality:**

3

**Strengths And Weaknesses:**

Strengths:

1. Advancing interpretability in multimodal sensor fusion is highly valuable, particularly for safety-critical applications such as autonomous driving.

2. The proposed method is well-grounded in theory, building on established frameworks like Layer-Wise Relevance Propagation and Deep Taylor Decomposition, and shows promising results in modality attribution.

Weaknesses:

1. The current evaluation and demonstrations appear limited to two-modality settings, without experiments or evidence showing its scalability to scenarios involving three or more modalities. While Appendix F is said to discuss generalization to >2 modalities, no experiments are shown for such settings. It’s unclear how well LMD scales or handles interference when more than two modalities are used.

2. While the paper discusses bias components extensively, the semantic meaning of bias remains underexplored (Please see question 3 below).

3. The paper doesn’t compare against other post-hoc multimodal interpretability methods, such as SHAP[1] or attention-based attribution techniques in transformer-based models [2], which are frequently adopted in multimodal fusion settings (e.g., [3])

[1] A unified approach to interpreting model predictions, NeruIPS 2017

[2] Transformer Interpretability Beyond Attention Visualization, CVPR 2021

[3] TransFuser: Imitation with Transformer-Based Sensor Fusion for Autonomous Driving, TPAMI 2023.

---

> ### Author Rebuttal · Authors · 2025-07-31
>
> We appreciate your thoughtful feedback.
> Your recognition of the value of interpretability for safety-critical fusion models and the theoretical soundness of LMD is encouraging. Below we address each concern in turn and outline changes we will include in the revised submission.
>
> ## **W1 & Q1**
> > **&nbsp;The current evaluation and demonstrations appear limited to two-modality settings, without experiments or evidence showing its scalability to scenarios involving three or more modalities**
>
> To address the concerns, we have extended LMD to a three-sensor setting (camera + radar + LiDAR) and performed a comprehensive variant study on bias-splitting strategies. This experiment is expected to further demonstrate the generality of LMD. For brevity, we share the correlation results using the same metrics as in the paper. To ensure reproducibility and enable other researchers to verify, we plan to release the code for all experiments presented.
>
>
>
>
> -------------------------------------------------------------------------------------------------------------------------------------------------------------------------------
> **Table A.** Three-modality ablation of LMD versus baselines
> | **Method** | **Act.** | **Norm.** | **CₚRₚ/L (↑)** | **CₚRₚ/C (↓)** | **CₚRₚ/R (↓)** | **CₚLₚ/R (↑)** | **CₚLₚ/C (↓)** | **CₚLₚ/L (↓)** | **RₚLₚ/C (↑)** | **RₚLₚ/R (↓)** | **RₚLₚ/ L(↓)** |
> |---------------------|:--------:|:---------:|:--------------:|:--------------:|:--------------:|:--------------:|:--------------:|:--------------:|:--------------:|:--------------:|:--------------:|
> | Baseline 1 | ✗ | ✗ | 0.3606 ± 0.095 | 0.0679 ± 0.069 | 0.1103 ± 0.095 | 0.1957 ± 0.092 | 0.0955 ± 0.069 | 0.2265 ± 0.094 | 0.6794 ± 0.099 | 0.4755 ± 0.092 | 0.4663 ± 0.104 |
> | Baseline 2 | ✓ | ✗ | 0.9764 ± 0.018 | 0.5459 ± 0.118 | 0.6565 ± 0.134 | 0.9717 ± 0.029 | 0.5488 ± 0.117 | 0.5360 ± 0.103 | 0.9865 ± 0.009 | 0.6722 ± 0.107 | 0.5384 ± 0.092 |
> | Baseline 3 | ✗ | ✓ | 0.3685 ± 0.089 | 0.0669 ± 0.072 | 0.1245 ± 0.086 | 0.2165 ± 0.084 | 0.0570 ± 0.073 | 0.2289 ± 0.098 | 0.6880 ± 0.098 | 0.4751 ± 0.092 | 0.4678 ± 0.104 |
> | **LMD (Ratio Rule)** | **✓** | **✓** | **1.0000 ± 0.000** | **0.1385 ± 0.232** | **0.0279 ± 0.042** | **1.0000 ± 0.000** | **0.1488 ± 0.132** | **0.2310 ± 0.057** | **1.0000 ± 0.000** |  **0.0373 ± 0.091** | **0.2023 ± 0.097** |
>
> ----------------------------------------------------------------------------------------------------------------------------------------------------------------
>
> **Table B.** Three-modality ablation of LMD variants
>   | **Method**            | **CₚRₚ/L (↑)** | **CₚRₚ/C (↓)** | **CₚRₚ/R (↓)** | **CₚLₚ/R (↑)** | **CₚLₚ/C (↓)** | **CₚLₚ/L (↓)** | **RₚLₚ/C (↑)** | **RₚLₚ/R (↓)** | **RₚLₚ/L (↓)** |
> |-----------------------|:--------------:|:--------------:|:--------------:|:--------------:|:--------------:|:--------------:|:--------------:|:--------------:|:--------------:|
> | Uniform - Identity    | 1.0000 ± 0.000 | 0.3224 ± 0.148 | 0.5572 ± 0.114 | 1.0000 ± 0.000 | 0.3164 ± 0.168 | 0.3162 ± 0.043 | 1.0000 ± 0.000 | 0.5572 ± 0.114 | 0.3322 ± 0.089 |
> | Identity - Identity   | 1.0000 ± 0.000 | 0.3315 ± 0.170 | 0.0971 ± 0.056 | 1.0000 ± 0.000 | 0.3305 ± 0.161 | 0.0039 ± 0.139 | 1.0000 ± 0.000 | 0.0926 ± 0.080 | 0.0042 ± 0.113 |
> | Identity - Uniform    | 0.9994 ± 0.008 | 0.3332 ± 0.160 | 0.0807 ± 0.008 | 0.9903 ± 0.079 | 0.3182 ± 0.145 | 0.0087 ± 0.1125 | 0.9999 ± 0.000 | 0.0870 ± 0.078 | 0.0085 ± 0.113 |
> | **Identity - Ratio** | **1.0000 ± 0.000** | **0.1385 ± 0.232** | **0.0279 ± 0.042** | **1.0000 ± 0.000** | **0.1488 ± 0.132** | **0.2310 ± 0.057** | **1.0000 ± 0.000** |  **0.0373 ± 0.091** | **0.2023 ± 0.097** |
>
> --------------------------------------------------------------------------------------------------------------------------------------------------------------------------
>
> > **&nbsp; It remains unclear whether any practical challenges arise in such cases, for example, in the selection of reference points or maintaining modality separation when scaling beyond two modalities.**
>
> We found LMD maintains Eq. (7) in the main paper (all modality-specific outputs sum exactly to the original prediction) while  preserving the separation property when extending to three modalities. Importantly, we did not observe any instability and used the same reference points selection strategy as in the two-modal case without modification.
> Since this concern was raised by multiple reviewers, we will incorporate these results into the camera-ready version of the paper.
>
>
> ## **W2, Q3** :
> > **&nbsp; Figure 2 shows "perceptual capability" from the bias term, which is described as representing constant offsets. What exactly does this component capture, residual cross-modal interactions, constant activations, or something else? If it truly represents constant information, does the bias-based prediction remain stable or show minor variations when presented with similar inputs?**
>
> This inquiry touches on a core component of our method. To summarize, the bias-based prediction refers to the model's perceptual output generated solely from the constant internal values—specifically, the bias terms—without using any sensor input (e.g., RGB values or Radar RCS values). This is tightly aligned with the central philosophy of LMD, which aims to interpret the model perception in a modality specific manner.
>
> The natural follow-up is : why does this constant bias term exhibit perceptual capability? The perceptual capability of bias term is largely originate from the linearized activation layers. The key lies in how activation layers are handled during linearization. When we linearize these layers, we store information about the activated neurons to reproduce the original model behavior under perturbed inputs. Then, at inference time, instead of using the activation function directly, the model uses the pre-stored information. As a result, the internal constant components (i.e., bias terms) follow these pre-determined activation patterns, which in turn gives rise to perceptual behavior. On the other hand, normalization layers, as discussed in the paper, are responsible only for scaling and thus we hypothesize that they are unlikely to contribute significantly to perceptual expressivity in the bias term.
>
> In rigorous perspective, as you correctly identified, the bias term captures cross-modal interactions, since the activation status of a neuron at certain location itself is determined largely by the modality-specific information.
>
> Accurately attributing these effects to specific modality contributions is a highly complex, with existing approaches such as Shapley-based methods known to face limitations in such contexts. This observation also relates to the limitation we acknowledged in the paper. We have partially addressed the limitation by integrating our approach with an explanatory model, which offers a complementary perspective. Details can be found in our response to ```Reviewer KNAK```.
>
>
>
> > **&nbsp; While the paper discusses bias components extensively, the semantic meaning of bias remains underexplored**
>
> We recognize that this can be regarded as a potential weakness, and we plan to elaborate on this point in the camera-ready version by providing a more detailed explanation and discussion.
>
>
> ## **W3, Q2** :
> > **&nbsp; The paper doesn’t compare against other post-hoc multimodal interpretability methods, such as SHAP or attention-based attribution techniques in transformer-based models, which are frequently adopted in multimodal fusion settings**
>
> In response to similar requests from multiple reviewers, we have conducted additional experiments involving SHAP [1] as well as attention-based attribution techniques in transformer-based models. For a comparison with SHAP or LRP, we suggest to check our response to ```Reviewer KNAK``` and ```Reviewer opMu```.
>
> > **&nbsp; Can the proposed method be readily applied to more complex fusion architectures, such as those incorporating cross-modal attention mechanisms — for example, the network used in TransFuser?**
>
> We have carefully considered the applicability of our method to more complex fusion architectures, such as those incorporating cross-modal attention mechanisms, including the TransFuser. While TransFuser is widely-referenced fusion architecture, we chose to conduct our additional experiments on CRN [2], a more recent model that operates on the nuScenes dataset. Additionally, results on the model —incorporating transformer-based architectures—can be found in our response to ```Reviewer 3nmi```.
>
> ----------------------------------------------------------------------------------
> [1] Lundberg, Scott M., and Su-In Lee. "A unified approach to interpreting model predictions." Advances in neural information processing systems 30 (2017).
>
> [2] Kim, Youngseok, et al. "Crn: Camera radar net for accurate, robust, efficient 3d perception." Proceedings of the IEEE/CVF International Conference on Computer Vision. 2023.

---

> ### Comment · Reviewer_UztF · 2025-08-03
> **Thank You for the Rebuttal and Clarifications**
>
> I thank the authors for the detailed response. I particularly appreciate the extensive experiments on the three-sensor setting and CRN. I believe this additional evidence demonstrates the practical value of the method, which could serve as a hands-on tool for researchers to investigate the internal properties of multi-modal fusion models. Moreover, the comparison with SHAP further enriches the analysis and highlights the advantages of LMD. In light of these improvements, I have decided to raise my score and recommend the paper for acceptance.
>
> One minor question: may I ask which dataset the camera + radar + LiDAR data is sourced from? Is it still based on nuScenes? Just curious.

---

> > ### Author Response · Authors · 2025-08-04
> >
> > We sincerely thank you for thoughtful review of our work and for raising your score after considering the additional experiments and analyses.
> >
> > Regarding your question, the 3 modalities experiments were conducted on the **nuScenes dataset**, consistent with our other demonstrations. If you have any further questions, we would be happy to address them.
> >
> > Once again, we truly appreciate your thoughtful feedback and support in recommending our paper for acceptance.

---

### Decision · Program_Chairs · 2025-09-17

**Decision:**

Accept (poster)

**Comment:**

The paper makes a solid contribution to an important problem in autonomous driving safety. Three reviewers explicitly raised their scores after the rebuttal, satisfied with the additional evidence. The comprehensive experiments, theoretical grounding, and practical applicability outweigh the remaining concern about >3 modality scalability. The work establishes a valuable baseline for interpretable multimodal fusion that the community can build upon.